# Reasoning of Large Language Models over Knowledge Graphs with Super-Relations

**Song Wang**[1][†], **Junhong Lin**[2], **Xiaojie Guo**[3], **Julian Shun**[2], **Jundong Li**[1], **Yada Zhu**[3]
[1]University of Virginia, [2]MIT CSAIL, [3]IBM
{sw3wv,jundong}@virginia.edu  {junhong,jshun}@mit.edu  {xiaojie.guo,yzhu}@ibm.com

## Abstract

While large language models (LLMs) have made significant progress in processing and reasoning over knowledge graphs, current methods suffer from a high non-retrieval rate. This limitation reduces the accuracy of answering questions based on these graphs. Our analysis reveals that the combination of greedy search and forward reasoning is a major contributor to this issue. To overcome these challenges, we introduce the concept of super-relations, which enables both forward and backward reasoning by summarizing and connecting various relational paths within the graph. This holistic approach not only expands the search space, but also significantly improves retrieval efficiency. In this paper, we propose the **ReKnoS** framework, which aims to **Re**ason over **Kno**wledge Graphs with **S**uper-Relations. Our framework's key advantages include the inclusion of multiple relation paths through super-relations, enhanced forward and backward reasoning capabilities, and increased efficiency in querying LLMs. These enhancements collectively lead to a substantial improvement in the successful retrieval rate and overall reasoning performance. We conduct extensive experiments on nine real-world datasets to evaluate ReKnoS, and the results demonstrate the superior performance of ReKnoS over existing state-of-the-art baselines, with an average accuracy gain of 2.92%.

## 1 Introduction

Recent advances in large language models (LLMs) have greatly improved their ability to perform reasoning in various natural language processing tasks (Brown et al., 2020; Wei et al., 2022). However, for more complex and knowledge-intensive reasoning tasks, such as open-domain question answering (Gu et al., 2021), the knowledge stored in LLM parameters alone can be insufficient for effectively performing these tasks (Li et al., 2024). To address this limitation, recent research has explored the integration of knowledge graphs (KGs), which represent structured information as entities and relationships, to provide external knowledge that can assist LLMs in answering such questions (Sun et al., 2023; Ma et al., 2024; Xu et al., 2024; Zhang et al., 2022).

Despite this progress, effectively leveraging knowledge graphs for question answering remains challenging due to the vast amount of information in KGs and their complex structures. Existing approaches typically resort to two retrieval strategies: (1) *subgraph-based reasoning* (Zhang et al., 2022; Jiang et al., 2023d), which retrieves subgraphs containing task-relevant triples from the KG and uses them as input for LLMs; and (2) *LLM-based reasoning* (Jiang et al., 2023b; Sun et al., 2023), which involves iteratively querying LLMs to select relevant paths within KGs. While subgraph-based reasoning can efficiently retrieve structured knowledge, LLM-based reasoning benefits from the comprehension capabilities of LLMs to precisely select relevant triples, resulting in superior performance compared to subgraph-based methods (Jiang et al., 2024; Pan et al., 2024).

However, even with the advances in LLM-based reasoning methods, there remains a significant performance gap that is often overlooked by existing research. Specifically, current approaches exhibit a substantial non-retrieval rate, indicating that in certain cases, the LLMs fail to find a satisfactory reasoning path within the maximum allowable reasoning length. As demonstrated in Figure 1, for the recent method ToG (Sun et al., 2023) on the prevalent dataset GrailQA (Gu et al., 2021), the accuracy reduces by around 10% when the information is not retrieved from the KG.

---

*† Work done when Song Wang was visiting MIT CSAIL and the MIT-IBM AI Watson Lab.

In this paper, we investigate the reasons behind retrieval failures when searching the knowledge graph, identifying two relation patterns that are often overlooked by existing works. As illustrated in Figure 2, the primary causes of non-retrieval can be classified into three categories: (1) **Misdirection**, where the correct path runs parallel to the retrieved path; (2) **Depth Limitation**, indicating that the correct path is longer than the retrieved one due to the maximum allowable length being exceeded; and (3) **Path Absence**, where the correct path does not exist in the knowledge graph. Notably, the first two cases account for the majority

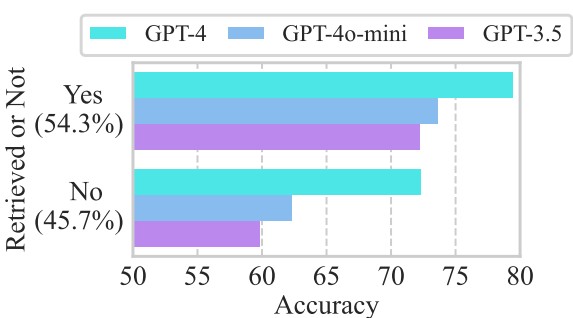

Figure 1: The accuracy (%) of retrieved and non-retrieved samples on the GrailQA dataset.

of non-retrieval instances. Existing methods struggle with the first case because they may initially select relations that appear promising but are ultimately incorrect. In such cases, once the first step is wrong, subsequent retrieval attempts are unlikely to capture the correct path.

To address the issue of misdirection, we propose a novel framework **ReKnoS**, which aims to **Re**ason over **Kno**wledge Graphs with **S**uper-Relations. *Super-relations* are defined as a group of relations within a specific field. For example, the relation "developer of" and "designer of" are both under the super-relation "video_game", as they are in the same field of video games (shown at the bottom of Figure 3). Using super-relations allows for a more holistic approach by enhancing both the width (the number of relations covered at each step) and depth (the total length of the reasoning path). By integrating super-relations, our framework can introduce additional relational paths that were previously inaccessible, thus expanding the search space and improving retrieval rates. These super-relations effectively summarize and connect disparate parts of the graph, facilitating a more comprehensive exploration of the data. Moreover, with the summarized information within super-relations, we propose to leverage it to achieve forward reasoning and backward reasoning, which aim at exploring future paths and previous alternative paths, respectively. With the design of super-relations, our framework can represent multiple relation paths with a

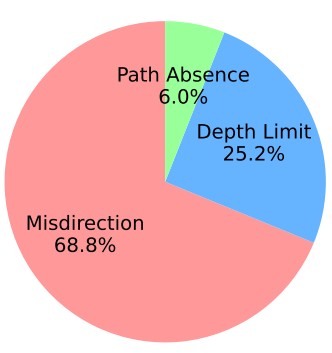

Figure 2: The non-retrieval cases on the GrailQA dataset with baseline ToG (Sun et al., 2023) (maximum length of 3).

super-relation during reasoning. As such, we do not need to abandon a large number of paths during reasoning, thus significantly increasing the search space and improving the retrieval rate.

The contributions of this paper are as follows:

- **Design:** The use of super-relations, defined as groups of relations within a specific field, enables the inclusion of a large number of relations for efficient reasoning over knowledge graphs. Moreover, super-relations can both enhance the depth and width of various reasoning paths to deal with non-retrieval.

- **Framework:** We propose a novel reasoning framework ReKnoS that integrates super-relations, enabling the representation of multiple relation paths simultaneously. This approach allows for a significant expansion of the search space by incorporating diverse relational paths without discarding potentially valuable connections during reasoning. Our code is provided at https://github.com/SongW-SW/REKNOS.

- **Experiments and Results:** We perform experiments on nine real-world datasets and show that our approach significantly outperforms traditional reasoning methods in both retrieval success rate and search space size, improving them by an average of 25% and 87%, respectively. These results highlight the effectiveness of our design choices in addressing complex knowledge graph reasoning tasks.

## 2 RELATED WORK

Although LLMs have exhibited expressive capabilities in various natural language process tasks, such as question answering tasks (Wei et al., 2022; Wang et al., 2023), they can lack up-to-date or domain-specific knowledge (Wang et al., 2023; Jiang et al., 2023b). Recently, researchers have proposed using knowledge graphs (KGs), which provide up-to-date and structured domain-specific information, to enhance the performance of LLMs (Pan et al., 2024). As a classic example, retrieval-augmented generation (RAG) methods have been widely used to incorporate external knowledge sources as additional input to LLMs (Jiang et al., 2023e). Nevertheless, the complex structures in KGs render such retrieval more difficult and may involve unnecessary noise (Petroni et al., 2021).

Knowledge graph question answering (KGQA) focuses on answering natural language questions using structured facts from knowledge graphs (Pan et al., 2024). A central challenge lies in efficiently retrieving relevant knowledge and applying it to derive accurate answers. Recent approaches to KGQA can be divided into two main categories: (1) subgraph-based reasoning and (2) LLM-based reasoning. Subgraph-based methods, such as SR (Zhang et al., 2022), UniKGQA (Jiang et al., 2023d), and ReasoningLM (Jiang et al., 2023c), perform reasoning over retrieved subgraphs that contain task-related fact triples from the KG. However, these methods are highly dependent on the quality of the retrieved subgraphs, which may contain irrelevant information and struggle to generalize to other structures within the KG. More recently, researchers have turned to LLMs to enhance reasoning on KGs. As an early effort, StructGPT (Jiang et al., 2023b) leverages LLMs to generate executable SQL queries to reason on structured data, including databases and KGs. Think-on-Graph (ToG) (Sun et al., 2023) extends this by using LLMs to select relevant relations and entities for retrieving target answers. KG-Agent (Jiang et al., 2024) further advances this line of work by incorporating a multifunctional toolbox that dynamically selects tools to explore and reason over KGs.

## 3 PRELIMINARIES

### 3.1 KNOWLEDGE GRAPH (KG)

A *knowledge graph* is composed of a large set of fact triples, represented as $G = \{\langle e, r, e' \rangle \mid e, e' \in E, r \in R\}$, where $E$ and $R$ denote the sets of entities and relations, respectively. Each triple $\langle e, r, e' \rangle$ captures a factual relationship, indicating that a relation $r$ exists between a head entity $e$ and a tail entity $e'$. For the KGQA task studied in this work, we assume the availability of a KG that contains the entities relevant to answering the given natural language question. Our goal is to design an LLM-based framework capable of performing reasoning on the KG to retrieve the answer to the question.

### 3.2 SUPER-RELATIONS

In this paper, we introduce the concept of a super-relation to efficiently gather information from a set of more granular, detailed relations. For instance, consider a super-relation "**music.featured_artist**," which encompasses a variety of specific relations, such as "*music.featured_artist.recordings*." Iterating over each of these detailed relations can be computationally intensive. By using a higher-level super-relation like "**music.featured_artist**," we abstract and group these related relations together, allowing us to query LLMs using a single, more general relation. This abstraction reduces the complexity of handling numerous fine-grained relations while preserving the relevant information needed for reasoning. In the prevalent knowledge graph Wikidata (Vrandečić & Krötzsch, 2014) that we use in this paper, relations are typically organized into a three-level hierarchy, such as "music," "featured_artist," and "recordings." We treat the second-level relations as super-relations, which serve as a representative for related third-level relations. However, it is important to note that not all KGs are structured with such clear hierarchical levels. In cases where a KG lacks this inherent structure, an alternative approach

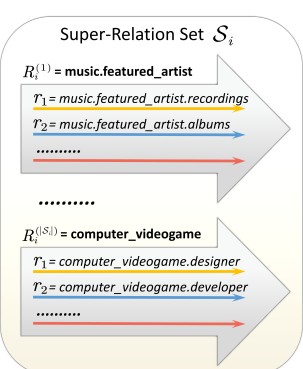

Figure 3: An example of various super-relations $R$ included in a super-relation set $\mathcal{S}$.

clear hierarchical levels. In cases where a KG lacks this inherent structure, an alternative approach

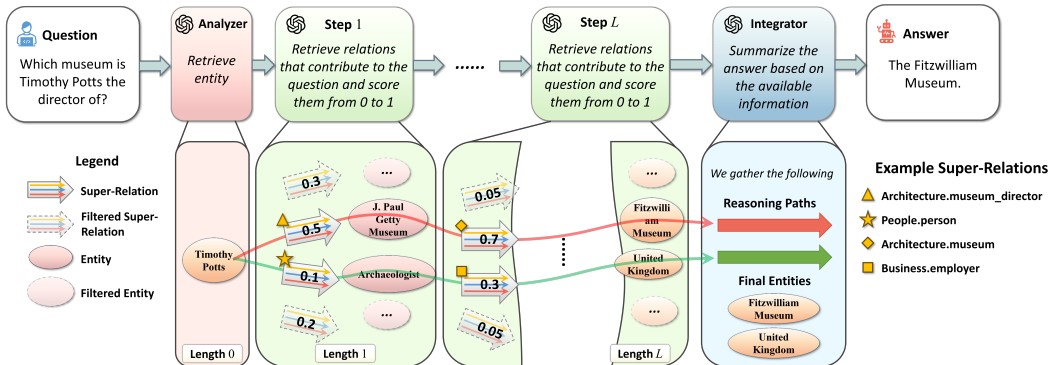

Figure 4: Overall ReKnoS framework. The LLM first extracts the query entity from the input question and then performs up to $L$ steps of reasoning. In each step, the LLM retrieves several super-relations and scores them. Only the selected candidates will be used for further reasoning. Finally, the LLM gathers the reasoning paths and final entities to generate the final answer.

involves clustering relations based on their semantic similarity and using the cluster centers as super-relations. These cluster centers must be textually coherent and meaningful to ensure they can effectively summarize the detailed relations within each cluster. In our experiments, we directly utilize the hierarchical levels.

We formally define a *super-relation* $R$ as a group of more granular, detailed relations $R = \{r_1, r_2, \ldots, r_n\}$, where each $r_i$ is a specific relation on the knowledge graph. Examples are shown Figure 3. For example, if $R$ is the super-relation "music.featured_artist," it might include a set of relations such as $R = \{$"music.featured_artist.recordings", "music.featured_artist.albums", $\ldots\}$.

### 3.3 SUPER-RELATION SEARCH PATHS

We first define a super-relation connection between two super-relations.

**Definition 1. *Super-Relation Connection.*** *Let $R_1$ and $R_2$ be two super-relations, each consisting of multiple relations. We define $R_1 \rightarrow R_2$ (i.e., $R_1$ connects to $R_2$) if and only if there exist entities $e_1$, $e_2$, and $e_3$ such that $e_1 \xrightarrow{r_1} e_2 \xrightarrow{r_2} e_3$, where $r_1 \in R_1$ and $r_2 \in R_2$.*

In this manner, based on connections between super-relations, we define a super-relation path and then define a super-relation search path, where each position in the path consists of multiple super-relations.

**Definition 2. *Super-Relation Path.*** *A super-relation path $P = (R_1 \rightarrow R_2 \rightarrow \ldots \rightarrow R_l)$ of length $l$ consist of $l$ super-relations that are consecutively connected.*

**Definition 3. *Super-Relation Search Path.*** *A super-relation search path $\mathcal{Q}_l$ of length $l$ consists of $l$ super-relation sets, i.e., $\mathcal{Q}_l = \{\mathcal{S}_1, \mathcal{S}_2, \ldots, \mathcal{S}_l\}$. Here $\mathcal{S}_i = \{R_i^{(1)}, R_i^{(2)}, \ldots, R_i^{(|\mathcal{S}_i|)}\}$ is a super-relation set at the $i$-th position in $\mathcal{Q}_l$. Moreover, in each super-relation set $\mathcal{S}_i$, there exists at least one super-relation that is connected to a super-relation in the subsequent set $\mathcal{S}_{i+1}$, i.e.,*

$$\forall R \in \mathcal{S}_i, \ \exists R' \in \mathcal{S}_{i+1} \text{ such that } R \rightarrow R', \ i = 1, 2, \ldots, l-1.$$

## 4 SUPER-RELATION REASONING

Our ReKnoS framework consists of at most $L$ reasoning steps, which is also the maximum length of search paths considered. During each reasoning step, we aim to retrieve a set of $N$ important super-relations that are the most semantically related to the query based on the selected super-relations from previous reasoning steps. In other words, at the beginning of the $l$-th reasoning step, the search path $\mathcal{Q}_{l-1} = \{\mathcal{S}_1, \mathcal{S}_2, \ldots, \mathcal{S}_{l-1}\}$ is of length $l-1$, i.e., containing $l-1$ sets of important super-relations. These super-relation sets are consecutively connected according to Definition 3, i.e., $\forall R \in \mathcal{S}_i, \exists R' \in \mathcal{S}_{i+1}$ such that $R \rightarrow R'$. Our goal in this reasoning step is to select an important set of $N$ super-relations in $\mathcal{S}_l$, while satisfying the condition that $\forall R \in \mathcal{S}_l, \exists R' \in \mathcal{S}_{l+1}$ such that $R \rightarrow R'$. Moreover, $|\mathcal{S}_i| = N$ for $i = 1, 2, \ldots, l$. At the end of each reasoning step, the LLM will be

asked to decide whether to extract the answer from the entities or to proceed to the next reasoning step. The overall framework is illustrated in Figure 4.

## 4.1 CANDIDATE SELECTION

To select the current $\mathcal{S}_l$, we first need to extract any candidate super-relation $R'$ that satisfies the connection requirement: $\forall R \in \mathcal{S}_{l-1}, \exists R' \in \mathcal{S}_l$ such that $R \to R'$. This means that $R'$ is connected by any super-relation in the last super-relation set $\mathcal{S}_{l-1}$. Therefore, we can represent the set of candidate super-relations $\mathcal{C}_l$ as

$$\mathcal{C}_l = \{R \mid \exists R' \in \mathcal{S}_{l-1} \text{ such that } R' \to R\}, \tag{1}$$

where $\mathcal{S}_{l-1}$ is the previous super-relation set in the search path $\mathcal{Q}_{l-1}$. This approach obtains all super-relations directly connected to the search path. As the connections are generally becoming larger as the search proceeds, it is likely that $|\mathcal{C}_l| > |\mathcal{S}_{l-1}| = N$ for $l \geq 2$. Therefore, we need to perform reasoning to filter out irrelevant super-relations in $\mathcal{C}_l$ to obtain $\mathcal{S}_l \subseteq \mathcal{C}_l$.

## 4.2 SCORING

In the scoring step, we use an LLM to assign a score for each of the super-relations in $\mathcal{C}_l$. The scores are denoted as $\{s_l^1, s_l^2, \ldots, s_l^M\}$, where $M = |\mathcal{C}_l|$. Note that this step is only needed if $M > N$. Based on these scores, we aim to select $N$ super-relations with the largest scores. The scores are obtained as follows:

$$\{s_l^1, s_l^2, \ldots, s_l^M\} = \text{LLM}(\{R_C^1, R_C^2, \ldots, R_C^M\}), \text{ where } R_C^i \in \mathcal{C}_l, \ i = 1, 2, \ldots, M. \tag{2}$$

The prompt to the LLM is as follows, setting $N = 3$ as an example:

```
You need to select three relations from the following candidate relations, which are
the most helpful for answering the question.
Question:
Topic Entity:
Candidate Relations:
Reply with the relations you selected from these candidate relations:
```

Note that in the prompt, we do not explicitly inform the LLM about the super-relations in order to reduce unnecessary complexity. Following ToG, we use a human-crafted example as additional input to LLMs to guide the scoring process. Based on these scores, we select the $N$ super-relations from $\mathcal{C}_l$ with the highest scores. We first sort the scores in descending order:

$$s_l^{(1)} \geq s_l^{(2)} \geq \cdots \geq s_l^{(N)} \geq \cdots \geq s_l^{(M)}, \tag{3}$$

where $s_l^{(i)}$ is the $i$-th largest score, and the corresponding super-relation is denoted as $R_l^{(i)}$.

The $N$ selected super-relations are denoted as follows:

$$\mathcal{S}_l = \{R_l^{(1)}, R_l^{(2)}, \ldots, R_l^{(N)}\}. \tag{4}$$

Then, we normalize the scores of the selected super-relations so that they sum to 1:

$$\bar{s}_l^{(k)} = \frac{s_l^{(k)}}{\sum_{j=1}^{N} s_l^{(j)}} \quad \text{for } k = 1, 2, \ldots, N. \tag{5}$$

In this way, we ensure that only the most relevant super-relations with the highest scores are considered in the next iteration, and their influence is appropriately scaled through normalization. The normalized scores $\{\bar{s}_l^{(1)}, \bar{s}_l^{(2)}, \ldots, \bar{s}_l^{(N)}\}$ will be later used for relevant path selection in the reasoning process.

## 4.3 SCORE-BASED ENTITY EXTRACTION SELECTION

Now we have obtained the selected super-relation search path of length $l$, i.e., $\mathcal{Q}_l = \{\mathcal{S}_1, \mathcal{S}_2, \ldots, \mathcal{S}_l\}$. Here, each super-relation set $\mathcal{S}_i$ consists of $N$ super-relations, i.e., $\mathcal{S}_i = \{R_i^{(1)}, R_i^{(2)}, \ldots, R_i^{(N)}\}$.

Moreover, each super-relation $R_i^{(j)}$ is associated with a score assigned during the scoring step, denoted as $\bar{s}_i^{(j)}$. At this point, we need to either use the retrieved super-relations to answer the question or proceed to further increase the length of the super-relation search path.

To find the potential answers and also provide sufficient information for the LLM to decide whether to continue retrieving the super-relation search path, we propose to extract the entities at the end of $\mathcal{Q}_l$, i.e., entities connected by super-relations at the last step of that path. These entities will be used to prompt the LLM to decide whether the retrieved entities are sufficient for answering the query. If the LLM responds that these entities are sufficient for answering the query, we will further prompt it for the answer. On the other hand, if the LLM suggests further exploring the reasoning, we will repeat the steps described above to retrieve a longer super-relation search path $\mathcal{Q}_{l+1}$. When $l = L$, we will always ask the LLM for the answer.

**Super-Relation Path Selection.** Due to the large number of entities connected by super-relations in $\mathcal{Q}_l$, we first identify $K$ most relevant super-relation paths from $\mathcal{Q}_l$. Particularly, we represent a super-relation path $P_i$ derived from $\mathcal{Q}_l$ as follows:

$$P_i = \left( R_1^{(k_{i,1})} \to R_2^{(k_{i,2})} \to \ldots \to R_l^{(k_{i,l})} \right), \text{ where } R_j^{(k_{i,j})} \to R_{j+1}^{(k_{i,j+1})}, \ j = 1, 2, \ldots, l-1. \quad (6)$$

Here, $k_{i,j} \in [1, |\mathcal{S}_j|]$ represents the index of the specific super-relation in the $j$-th position of the $i$-th path, such that $R_j^{(k_{i,j})} \in \mathcal{S}_j$. The set of all possible super-relation paths is denoted as $\mathcal{P} = \{P_1, P_2, \ldots, P_{|\mathcal{P}|}\}$. Notably, the number of possible super-relation paths, i.e., $|\mathcal{P}|$, may vary depending on the connectivity across different super-relation sets in $P_l$. However, $\mathcal{P}$ contains at least one super-relation path, i.e., $|\mathcal{P}| \geq 1$ according to the following theorem.

**Theorem 4.1.** *A super-relation search path leads to at least one super-relation path. Given a selected super-relation search path of length l, $\mathcal{Q}_l = \{\mathcal{S}_1, \mathcal{S}_2, \ldots, \mathcal{S}_l\}$, there exists at least one super-relation path of length l, i.e., $P = \left( R_1^{(k_1)} \to R_2^{(k_2)} \to \ldots \to R_l^{(k_l)} \right)$, where $R_j^{(k_j)} \in \mathcal{S}_j$ and $k_j$ represents $k_{i,j}$ for a fixed i.*

Intuitively, this theorem holds because consecutive super-relation sets, i.e., $\mathcal{S}_i$ and $\mathcal{S}_{i+1}$, are connected, and thus at least one super-relation in each of them is connected to each other according to Definition 3. Therefore, the consecutively connected super-relations form a super-relation path. The proof is provided in Appendix A. To select the super-relation paths that are more relevant to the input query, we propose to sum the scores of the $l$ super-relations in each super-relation path. Specifically, for the $i$-th super-relation path $P_i = \left( R_1^{(k_{i,1})} \to R_2^{(k_{i,2})} \to \ldots \to R_l^{(k_{i,l})} \right)$, the total score is computed as

$$\text{Score}(P_i) = \sum_{j=1}^{l} \bar{s}_j^{(k_{i,j})} = \bar{s}_1^{(k_{i,1})} + \bar{s}_2^{(k_{i,2})} + \ldots + \bar{s}_l^{(k_{i,l})}. \quad (7)$$

With the summed scores for these super-relation paths, we select the $K$ super-relation paths with the largest scores. The final selected super-relation paths are as follows:

$$\mathcal{P}^* = \underset{\mathcal{P}_f \subseteq \mathcal{P} \, : \, |P_f| \leq K}{\arg\max} \sum_{P \in \mathcal{P}_f} \text{Score}(P). \quad (8)$$

**Final Entity Selection.** With the relevant super-relation paths, our final goal is to extract the entities at the end of each reasoning path in $\mathcal{P}^*$. We first introduce the following theorem for selecting specific relation paths from super-relation paths.

**Theorem 4.2.** *For any super-relation path $P = \left( R_1^{(k_1)} \to R_2^{(k_2)} \to \ldots \to R_l^{(k_l)} \right)$, there exists at least one corresponding KG reasoning path of length l, represented as $e_1 \xrightarrow{r_1} e_2 \xrightarrow{r_2} \cdots \xrightarrow{r_l} e_{l+1}$, where $r_i \in R_i^{(k_i)}$ is a relation chosen from the super-relation $R_i^{(k_i)}$, and $e_1, e_2, \ldots, e_{l+1}$ are entities in the knowledge graph.*

This theorem holds because in a super-relation path, according to Definition 2, two consecutive super-relations are connected, which leads to the connection of two relations. Therefore, the consecutively connected relations will form a relation path. The proof is provided in Appendix B. According to the

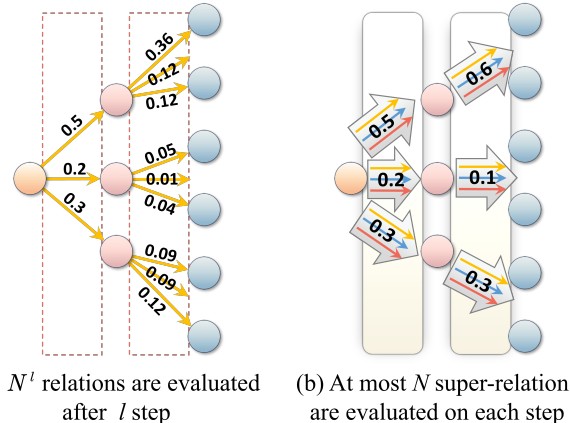

(a) $N^l$ relations are evaluated after $l$ step

(b) At most $N$ super-relations are evaluated on each step

Figure 5: Number of relations need to score by the LLM in (a) the baseline ToG (Sun et al., 2023) and (b) our framework ReKnoS. The numbers on the arrows represent LLM evaluated scores and correspond to LLM computations. For clarity, we omit some of the blue nodes.

theorem, we know that the selected super-relation path $P$ consists of at least one valid relation path in the knowledge graph. Therefore, this relation path can be used to extract entities that contribute to answering the query.

In Figure 5, we show that for the baseline ToG, the number of LLM calls grows exponentially as $l$ increases. However, for our framework ReKnoS, the required LLM calls at each step remain constant, indicating the efficiency of our framework. However, one challenge remains—there is a large number of entities covered by these super-relations, which can make it difficult to find the relevant ones. We describe how to address this issue below.

With these super-relation paths in $\mathcal{P}^*$, we aim to extract the entities that satisfy the connection property of these relations. The entity set for the entire super-relation path $P = \left( R_1^{(k_1)} \to R_2^{(k_2)} \to \ldots \to R_l^{(k_l)} \right)$ can be expressed as

$$E_l^{(k_l)} = \{e \mid \exists r_1 \in R_1^{(k_1)}, r_2 \in R_2^{(k_2)}, \ldots, r_l \in R_l^{(k_l)}, \text{ such that } e_1 \xrightarrow{r_1} e_2 \xrightarrow{r_2} \cdots \xrightarrow{r_l} e\}. \quad (9)$$

This set $E_l^{(k_l)}$ contains all of the final entities that satisfy the super-relation path $P$. Note that although each super-relation in $P$ can contain multiple relations, the size of $E_l^{(k_l)}$ should be much smaller than $\mathcal{P}^*$, as it requires $l$ entities on the super-relation path to be consecutively connected, which is a stricter requirement than having $l$ super-relations consecutively connected (see Definition 1).

We use the obtained entity set $E_l^{(k_l)}$ along with the super-relations in $P$ to query the LLM regarding the next step. The LLM will either output an inferred answer from $E_l^{(k_l)}$ or decide to continue the process by proceeding to the subsequent super-relations, i.e., repeating the process starting from Section 4.1 and increasing the super-relation path length $l$ by 1. Notably, when $l = L$, we will always ask the LLM for the answer.

## 5 EXPERIMENTS

**Baselines.** For baselines, we compare **ReKnoS** with several methods, including standard prompting (IO prompt) (Brown et al., 2020), Chain-of-Thought prompting (CoT prompt) (Wei et al., 2022), and Self-Consistency (Wang et al., 2023). Additionally, we include previous state-of-the-art (SOTA) approaches tailored for reasoning on knowledge graphs: StructGPT (Jiang et al., 2023b), Think-on-Graph (ToG) (Sun et al., 2023), and KG-Agent (Jiang et al., 2024).

**Datasets.** To evaluate the performance of our framework on multi-hop knowledge-intensive reasoning tasks, we conduct tests using four KBQA datasets: CWQ (Talmor & Berant, 2018), WebQSP (Yih et al., 2016), GrailQA (Gu et al., 2021), and SimpleQA (Bordes et al., 2015). Among these, three are multi-hop reasoning datasets with reasoning path lengths generally larger than one, and one is single-hop reasoning (i.e., SimpleQA) with reasoning paths of length one. Additionally, we include one

Table 1: The results (Hits@1 in %) of different methods on various Datasets, using GPT-3.5 and GPT-4o-mini as the LLM. The best results are highlighted in bold.

| Dataset | WebQSP | GrailQA | CWQ | SimpleQA | WebQ | T-REx | zsRE | Creak | Hotpot QA |
|---|---|---|---|---|---|---|---|---|---|
| LLM | | | | | **GPT-3.5** | | | | |
| **IO** | 63.3 | 29.4 | 37.6 | 20.0 | 48.7 | 33.6 | 27.7 | 89.7 | 28.9 |
| **Chain-of-Thought** | 62.2 | 28.1 | 38.8 | 20.3 | 48.5 | 32.0 | 28.8 | 90.1 | 34.4 |
| **Self-Consistency** | 61.1 | 29.6 | 45.4 | 18.9 | 50.3 | 41.8 | 45.4 | 90.8 | 35.4 |
| **ToG** | 76.2 | 68.7 | 57.1 | **53.6** | 54.5 | 76.4 | 88.0 | 91.2 | 35.3 |
| **KG-Agent** | 79.2 | 68.9 | 56.1 | 50.7 | 55.9 | **79.8** | 85.3 | 90.3 | 36.5 |
| **StructGPT** | 75.2 | 66.4 | 55.2 | 50.3 | 53.9 | 75.8 | 86.2 | 89.5 | 34.9 |
| **ReKnoS (Ours)** | **81.1** | **71.9** | **58.5** | 52.9 | **56.7** | 78.9 | **88.7** | **91.7** | **37.2** |
| LLM | | | | | **GPT-4o-mini** | | | | |
| **ToG** | 80.7 | 79.7 | 65.4 | 66.1 | 57.0 | 77.2 | 87.9 | 95.6 | 38.9 |
| **KG-Agent** | 81.2 | 77.5 | **67.0** | **67.9** | 56.2 | **80.6** | 86.5 | 96.1 | 40.1 |
| **StructGPT** | 79.5 | 78.2 | 64.7 | 63.0 | 54.5 | 76.6 | 88.1 | 94.9 | 38.5 |
| **ReKnoS (Ours)** | **83.8** | **80.5** | 66.8 | 67.2 | **57.6** | 78.5 | **88.4** | **96.7** | **40.8** |

open-domain QA dataset, WebQ (Berant et al., 2013), two slot-filling datasets, T-REx (Elsahar et al., 2018) and Zero-Shot RE (Petroni et al., 2021), one multi-hop complex QA dataset, HotpotQA (Yang et al., 2018), and one fact-checking dataset, Creak (Onoe et al., 2021). For the larger datasets, GrailQA and SimpleQA, 1,000 samples were randomly selected for testing to reduce computational overhead. For all of the datasets, we use exact match accuracy (Hits@1) as our evaluation metric, consistent with prior studies (Li et al., 2024; Jiang et al., 2023b). We use Freebase (Bollacker et al., 2008) as the KG for CWQ, WebQSP, GrailQA, SimpleQA, and WebQ. We use Wikidata (Vrandečić & Krötzsch, 2014) as the KG for T-REx, Zero-Shot RE, HotpotQA, and Creak. The implementation details are provided in Appendix C.

Due to space constraints, some experimental results and discussions are deferred to Appendix D.

## 5.1 COMPARATIVE RESULTS

The Hits@1 performance of our **ReKnoS** framework is evaluated on both GPT-3.5 and GPT-4o-mini across multiple datasets, as shown in Table 1. The results highlight several key insights. First, our framework consistently outperforms the baselines across most datasets, particularly in more complex datasets like **GrailQA** and **CWQ**. On **GrailQA**, for example, our framework achieves an accuracy of 71.9% on GPT-3.5 and 80.5% on GPT-4o-mini, compared to the best baseline (**KG-Agent**) with 68.9% and 77.5% accuracy, respectively. This improvement can be attributed to our model's more effective integration of reasoning mechanisms and structured knowledge representation, which are critical for answering more complex questions.

On simpler datasets, such as **SimpleQA** and **WebQ**, our approach also shows competitive performance. Although the margin of improvement is smaller compared to other baselines, our framework still surpasses them. For example, on GPT-3.5, our framework achieves 52.9% accuracy on **SimpleQA**, which is higher than **KG-Agent**'s 50.7% accuracy, demonstrating the robustness of our framework even on tasks that may not demand intricate reasoning chains.

Furthermore, we observe that the performance gap between our framework and the baselines becomes more pronounced when transitioning from GPT-3.5 to GPT-4o-mini. On **CWQ**, for instance, our framework improves from 58.5% (GPT-3.5) to 66.8% (GPT-4o-mini) accuracy, while the best-performing baseline (**KG-Agent**) improves from 56.1% to 67.0% accuracy. This suggests that our framework better leverages the increased reasoning and understanding capabilities of GPT-4o-mini, particularly in datasets that require multi-step reasoning, such as **CWQ** and **GrailQA**. Moreover, we also observe that the baseline KG-Agent performs better on SimpleQA and T-REx using the GPT-4o-mini LLM. This is potentially due to the advanced tool-using capabilities of GPT-4o-mini that can benefit KG-Agent, which performs reasoning over KGs with a specialized toolbox.

In conclusion, our results demonstrate the effectiveness and adaptability of our approach across a range of datasets, consistently outperforming state-of-the-art baselines. The transition to more

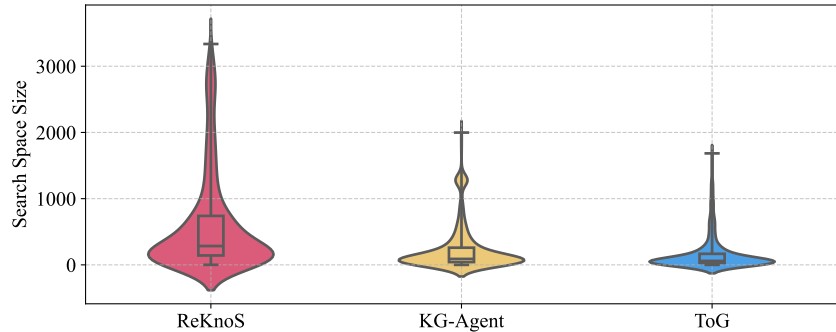

Figure 6: The size of the search space (i.e., the number of fact triples encountered during searching) of different methods on dataset GrailQA using GPT-3.5. The shape denotes the distribution of the search space sizes across all samples.

advanced language models like GPT-4o-mini further amplifies our model's strengths, especially in datasets that require deeper reasoning and knowledge inference.

## 5.2 EFFECT OF BACKBONE MODELS

In this subsection, we conduct experiments to compare the performance of **Re-KnoS** and the recent baseline ToG using different backbone models to study the impact of model parameter sizes. Notably, we consider the following LLMs in addition to the two LLMs used in Sec. 5.1: Llama-2-7B (Touvron et al., 2023), Mistral-7B (Jiang et al., 2023a), Llama-3-8B (Dubey et al., 2024), and GPT-4 (Anand et al., 2023). As seen in Table 2, the performance of **ReKnoS** varies depending on the choice of the underlying backbone model. First, smaller models, like Llama-2-7B, generally achieve worse results than larger models on all datasets. Nevertheless, the performance improvement of **ReKnoS** over ToG is more sig-

Table 2: Results (Hits@1 in %) of **ReKnoS** and ToG on various backbone models on four datasets.

| Model | WebQSP | GrailQA | CWQ | SimpleQA |
|---|---|---|---|---|
| **Llama-2-7B** | | | | |
| ToG | 61.2 | 60.3 | 49.8 | 35.7 |
| ReKnoS | 64.7 | 62.2 | 51.2 | 41.0 |
| **Mistral-2-7B** | | | | |
| ToG | 61.9 | 62.6 | 50.3 | 39.2 |
| ReKnoS | 62.5 | 64.3 | 53.1 | 45.3 |
| **Llama-3-8B** | | | | |
| ToG | 64.4 | 63.9 | 53.2 | 42.7 |
| ReKnoS | 67.9 | 65.8 | 56.7 | 46.4 |
| **GPT-4** | | | | |
| ToG | 82.6 | 81.4 | 67.6 | 66.7 |
| ReKnoS | 84.9 | 82.7 | 68.2 | 69.3 |

nificant on smaller models, compared to large models like GPT-4. This indicates that our framework is significantly better for smaller models that could be easily deployed, making our framework more practical. Moreover, our framework consistently outperforms ToG on models of various parameter sizes, indicating the advantage of our framework in helping smaller models solve complex tasks. Finally, GPT-4, as expected, provides the highest performance across all datasets. This further underscores the potential of more advanced models like GPT-4 to handle complex, multi-step reasoning tasks and diverse queries, outperforming smaller and less sophisticated models. Nevertheless, smaller models with higher efficiency are valuable in simpler tasks when the inference latency is important.

## 5.3 SUPER-RELATION ANALYSIS

In this subsection, we investigate how the incorporation of super-relations contributes to improved accuracy and running time. As an example, we present the size of the search space of different methods on dataset GrailQA in Figure 6. We observe that our method has the largest search space, with nearly 42% and 55% improvements in average size over the state-of-the-art methods KG-Agent and ToG, respectively. This indicates that with super-relations, we can significantly increase the search space size when we perform reasoning on a KG. As a consequence, our method can encounter more fact triples and is more likely to retrieve the correct one for answering the question.

## 5.4 EFFICIENCY ANALYSIS

LLM calls are the dominant part of the computation; we hence separately analyze LLM calls here. The LLM operations typically require the most time and computational resources in such frameworks, especially when compared to other processes like preprocessing or document retrieval.

Let $N$ and $L$ represent the search width and length, respectively. In each reasoning step, **ReKnoS** calls the LLM twice, where

Table 3: Comparison of the average number of calls, query times (in seconds), and Hits@1 on the WebQSP and GrailQA datasets using GPT-3.5.

| Model | WebQSP | | | GrailQA | | |
|---|---|---|---|---|---|---|
| | # Calls | Time | Hits@1 | # Calls | Time | Hits@1 |
| **ToG** | 15.3 | 4.9 | 76.2 | 18.9 | 6.8 | 68.7 |
| **StructGPT** | 11.0 | 2.0 | 75.2 | 13.3 | 3.5 | 66.4 |
| **ReKnoS** | 7.2 | 3.7 | 81.1 | 10.2 | 5.1 | 71.9 |

the first one is to score the super-relations, and the second one is to determine whether to proceed to the next reasoning step. Therefore, in this manner, **ReKnoS** calls the LLM $(2L^* + 1)$ times per question, where $L^*$ is the final number of reasoning steps used and the other one is to answer the final question. Note that the value of $L^*$ can be smaller than the value of $L$.

The significant advantage of **ReKnoS** lies in the fact that the number of calls is independent of the width $N$ as we utilize the concept of super-relations. In contrast, ToG searches for each path individually, and the resulting total number of LLM calls is upper bounded by $2LN+1$ (ToG calls the LLM twice at each reasoning step for each relation, while also requiring a final call for the answer). This reliance on $N$ significantly increases the number of LLM calls needed.

We show the number of LLM calls, average query times, and Hits@1 for the different methods in Table 3. As seen in the table, **ReKnoS** achieves the lowest average number of LLM calls while maintaining superior performance, underscoring both its efficiency and effectiveness. Although StructGPT is faster in terms of query times, it suffers from lower accuracy. This trade-off between speed and accuracy further highlights the balance **ReKnoS** achieves, as it maintains high accuracy without excessively increasing the number of LLM calls.

## 5.5 HYPER-PARAMETER ANALYSIS

Here, we evaluate how hyper-parameters affect the performance of **ReKnoS**. We present the results of our framework in Table 4 with different values of $L$ and $N$. From the results, we see that increasing both $L$ and $N$ leads to better performance. This is because, with larger values of $L$ and $N$, our framework can retrieve and store more information in the super-relation search path $\mathcal{Q}$, which can contain up to $NL$

Table 4: Accuracy and the retrieval rate (Ret.) of **ReKnoS** with different hyper-parameters using GPT-3.5.

| Setting | $N = 1$ | | $N = 3$ | | $N = 5$ | |
|---|---|---|---|---|---|---|
| | Hits@1 | Ret. | Hits@1 | Ret. | Hits@1 | Ret. |
| $L = 1$ | 75.3 | 57.0 | 79.2 | 69.2 | 79.8 | 71.3 |
| $L = 3$ | 76.2 | 64.3 | 81.1 | 69.8 | 81.8 | 76.6 |
| $L = 5$ | 77.3 | 68.6 | 82.1 | 72.9 | 82.2 | 76.9 |

super-relations. It is noteworthy that decreasing $N$ from 3 to 1 causes a considerable drop in accuracy. The reduction in Hits@1 is substantial. For example, when $L = 3$, Hits@1 drops from 81.1% with $N = 3$ to 76.2% with $N = 1$. This indicates that $N$ plays a crucial role in the framework's performance, as having fewer super-relations reduces the amount of information available for reasoning, leading to less effective answers.

## 6 CONCLUSION

This paper introduces the **ReKnoS** framework, which leverages super-relations to involve a large number of relations during reasoning in knowledge graphs. By enabling the representation and exploration of multiple relation paths simultaneously, our approach significantly expands the search space of reasoning paths over knowledge graphs without sacrificing potentially valuable information. Extensive experiments demonstrate the effectiveness of our method, showing that **ReKnoS** achieves substantial improvements over state-of-the-art baselines. These results underscore the potential of super-relations in advancing complex reasoning tasks in knowledge graph applications. In future work, we will apply our work to knowledge graphs of other domains and types.

ACKNOWLEDGEMENTS

This work is supported in part by the MIT-IBM AI Watson Lab, National Science Foundation (NSF) under grants CCF-1845763, CCF-2316235, and CCF-2403237, IIS-2006844, IIS-2144209, IIS-2223769, CNS-2154962, BCS-2228534, and CMMI-2411248; the Commonwealth Cyber Initiative (CCI) under grant VV-1Q24-011; Google Faculty Research Award; and Google Research Scholar Award.

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

## A    PROOF OF THEOREM 4.1

**Theorem 4.1.** *A super-relation search path leads to at least one super-relation path. Given a selected super-relation search path of length l, $\mathcal{Q}_l = \{\mathcal{S}_1, \mathcal{S}_2, \ldots, \mathcal{S}_l\}$, there exists at least one super-relation path of length l, i.e., $P = \left( R_1^{(k_1)} \to R_2^{(k_2)} \to \ldots \to R_l^{(k_l)} \right)$, where $R_j^{(k_j)} \in \mathcal{S}_j$ and $k_j$ represents $k_{i,j}$ for a fixed i.*

*Proof.* Let $\mathcal{Q}_l = \{\mathcal{S}_1, \mathcal{S}_2, \ldots, \mathcal{S}_l\}$ represent a super-relation search path consisting of $l$ super-relation sets, as per Definition 3. Each super-relation set $\mathcal{S}_i = \{R_i^{(1)}, R_i^{(2)}, \ldots, R_i^{(|\mathcal{S}_i|)}\}$ contains multiple super-relations.

According to Definition 3, in each super-relation set $\mathcal{S}_i$, there exists at least one super-relation $R \in \mathcal{S}_i$ that connects to a super-relation in the next set $\mathcal{S}_{i+1}$. In other words,

$$\forall R \in \mathcal{S}_i, \exists R' \in \mathcal{S}_{i+1} \text{ such that } R \to R'.$$

This ensures that there are connected super-relations between consecutive sets.

We prove the theorem using induction. As the base case, we observe that according to Definition 3, there exists a super-relation super-relation $R_1^{(k_1)} \in \mathcal{S}_1$ and another super-relation $R_2^{(k_2)} \in \mathcal{S}_2$ such that:

$$R_1^{(k_1)} \to R_2^{(k_2)}.$$

As the inductive hypothesis, suppose that for a super-relation search path of length $l - 1$, where $l > 2$, there exists at least one super-relation path of length $l - 1$. We take one such path $\left( R_1^{(k_1)} \to R_2^{(k_2)} \to \ldots \to R_{l-1}^{(k_{l-1})} \right)$, where $R_j^{(k_j)} \in \mathcal{S}_j$. We take the last super-relation on such

a path, $R_{l-1}^{(k_l-1)} \in \mathcal{S}_{l-1}$. By Definition 3, we know that there exists a connected super-relation $R_l^{(k_l)} \in \mathcal{S}_l$ such that:

$$R_{l-1}^{(k_l-1)} \to R_l^{(k_l)}.$$

Then, we can form the super-relation path $P = \left( R_1^{(k_1)} \to R_2^{(k_2)} \to \dots \to R_{l-1}^{(k_l-1)} \to R_l^{(k_l)} \right)$. Thus, we have that every super-relation search path leads to at least one super-relation path, as required.

$\square$

## B    PROOF OF THEOREM 4.2

**Theorem 4.2.** *For any super-relation path $P = \left( R_1^{(k_1)} \to R_2^{(k_2)} \to \dots \to R_l^{(k_l)} \right)$, there exists at least one corresponding KG reasoning path of length $l$, represented as $e_1 \xrightarrow{r_1} e_2 \xrightarrow{r_2} \cdots \xrightarrow{r_l} e_{l+1}$, where $r_i \in R_i^{(k_i)}$ is a relation chosen from the super-relation $R_i^{(k_i)}$, and $e_1, e_2, \dots, e_{l+1}$ are entities in the knowledge graph.*

*Proof.* Given a super-relation path $P = \left( R_1^{(k_1)} \to R_2^{(k_2)} \to \dots \to R_l^{(k_l)} \right)$, we need to show that there exists a corresponding reasoning path in the knowledge graph such that

$$e_1 \xrightarrow{r_1} e_2 \xrightarrow{r_2} \cdots \xrightarrow{r_l} e_{l+1},$$

where each $r_i$ is a relation from the super-relation $R_i^{(k_i)}$, and the entities $e_1, e_2, \dots, e_{l+1}$ are KG entities.

From Definition 1, we know that each super-relation $R_i$ is composed of multiple relations, i.e., $R_i = \{r_1, r_2, \dots\}$, where $r_j$ is a standard binary relation between two entities in the knowledge graph. The super-relation $R_1^{(k_1)}$ connects to $R_2^{(k_2)}$, denoted as $R_1^{(k_1)} \to R_2^{(k_2)}$, if there exist entities $e_1, e_2$, and $e_3$ such that

$$e_1 \xrightarrow{r_1} e_2 \xrightarrow{r_2} e_3,$$

where $r_1 \in R_1^{(k_1)}$ and $r_2 \in R_2^{(k_2)}$.

According to Definition 2, a super-relation path $P = \left( R_1^{(k_1)} \to R_2^{(k_2)} \to \dots \to R_l^{(k_l)} \right)$ consists of $l$ consecutive super-relations that are connected. This means that for each consecutive pair of super-relations $R_i^{(k_i)}$ and $R_{i+1}^{(k_{i+1})}$, there exist entities $e_i, e_{i+1}$, and $e_{i+2}$ such that

$$e_i \xrightarrow{r_i} e_{i+1} \xrightarrow{r_{i+1}} e_{i+2},$$

where $r_i \in R_i^{(k_i)}$ and $r_{i+1} \in R_{i+1}^{(k_{i+1})}$.

We proceed by induction on the length $l$ of the super-relation path.

Base Case ($l = 1$): For a super-relation path of length 1, $P = (R_1^{(k_1)})$, we have the super-relation $r_1 \in R_1^{(k_1)}$ and two entities $e_1$ and $e_2$ such that

$$e_1 \xrightarrow{r_1} e_2.$$

This forms a valid KG reasoning path of length 1.

Our inductive hypothesis is that for a super-relation path of length $l - 1$, $P = (R_1^{(k_1)} \to R_2^{(k_2)} \to \dots \to R_{l-1}^{(k_l-1)})$ (for $l > 2$), there exists a corresponding KG reasoning path

$$e_1 \xrightarrow{r_1} e_2 \xrightarrow{r_2} \cdots \xrightarrow{r_{l-1}} e_l,$$

where each $r_i \in R_i^{(k_i)}$ is chosen from the respective super-relation.

Table 5: Evaluation results (Part 1) with average accuracy and standard deviations over five runs.

| Model | WebQSP | GrailQA | CWQ | SimpleQA | WebQ |
|-------|--------|---------|-----|----------|------|
| ToG | $76.4 \pm 2.0$ | $68.9 \pm 1.5$ | $56.2 \pm 1.8$ | $53.7 \pm 0.9$ | $54.1 \pm 2.5$ |
| KG-Agent | $78.6 \pm 1.9$ | $69.4 \pm 2.1$ | $56.4 \pm 1.2$ | $52.5 \pm 2.7$ | $55.1 \pm 2.0$ |
| ReKnoS (Ours) | $\mathbf{81.0} \pm 1.0$ | $\mathbf{71.2} \pm 1.5$ | $\mathbf{57.7} \pm 0.9$ | $\mathbf{54.1} \pm 1.0$ | $\mathbf{56.5} \pm 1.2$ |

Table 6: Evaluation results (Part 2) with average accuracy and standard deviations over five runs.

| Model | T-REx | zsRE | Creak | Hotpot QA |
|-------|-------|------|-------|-----------|
| ToG | $76.0 \pm 1.3$ | $87.5 \pm 1.0$ | $91.3 \pm 2.8$ | $35.6 \pm 2.3$ |
| KG-Agent | $78.9 \pm 0.8$ | $86.8 \pm 1.5$ | $90.3 \pm 2.5$ | $36.5 \pm 1.1$ |
| ReKnoS (Ours) | $\mathbf{79.8} \pm 1.1$ | $\mathbf{88.3} \pm 0.7$ | $\mathbf{91.9} \pm 0.9$ | $\mathbf{37.1} \pm 1.4$ |

Now, consider a super-relation path of length $l$, $P = (R_1^{(k_1)} \to R_2^{(k_2)} \to \ldots \to R_{l-1}^{(k_{l-1})} \to R_l^{(k_l)})$. From Definition 1, the super-relation $R_{l-1}^{(k_{l-1})} \to R_l^{(k_l)}$ exists if and only if there exist entities $e_l$ and $e_{l+1}$ such that

$$e_l \xrightarrow{r_l} e_{l+1},$$

where $r_l \in R_l^{(k_l)}$.

Thus, the extended path is

$$e_1 \xrightarrow{r_1} e_2 \xrightarrow{r_2} \cdots \xrightarrow{r_{l-1}} e_l \xrightarrow{r_l} e_{l+1},$$

which forms a valid KG reasoning path of length $l$, as required.

$\square$

## C   IMPLEMENTATION DETAILS

Since our framework is flexible and can be applied to any LLMs, in our experiments, we mainly consider two large LLMs: GPT-3.5 (ChatGPT) (OpenAI, 2022) and GPT-4o-mini (Anand et al., 2023). We additionally consider Llama-2-7B (Touvron et al., 2023), Mistral-7B (Jiang et al., 2023a), Llama-3-8B (Dubey et al., 2024), and GPT-4 (Anand et al., 2023) in Sec. 5.2. We run all of our experiments on one NVIDIA A6000 GPU with 48GB of memory. Across all datasets and methods, we set the width $N$ to 3 and the maximum length $L$ to 3. When prompting the LLM to score super-relations, we use 3 examples as in-context learning demonstrations, following the existing work on ToG (Sun et al., 2023). The starting entities of each query are provided in the datasets.

## D   ADDITIONAL RESULTS AND DISCUSSIONS

### D.1   ADDITIONAL EVALUATION ANALYSIS

Following the settings of ToG and KG-Agent, we initially did not run multiple rounds of evaluation. To strengthen our analysis, we conducted five additional experimental runs and report the average results along with standard deviations. This ensures that our observed improvements are statistically significant. The results are shown in Table 5 and Table 6.

From these results, we observe slight variations across multiple rounds of experiments. However, **ReKnoS** consistently achieves the best performance compared to the two state-of-the-art baselines, highlighting its effectiveness and robustness.

### D.2   COMPARISON WITH SUBGRAPH-BASED REASONING METHODS

To further evaluate the effectiveness of our proposed **ReKnoS** framework, we conduct additional experiments comparing with subgraph-based reasoning methods **SR** (Zhang et al., 2022) and **UniKGQA** (Jiang et al., 2023d).

Table 7: Performance comparison of subgraph-based reasoning methods and ReKnoS on benchmark KGQA datasets.

| Model | WebQSP | GrailQA | CWQ | SimpleQA |
|---|---|---|---|---|
| SR | 68.9 | 65.2 | 50.2 | 48.7 |
| UniKGQA | 75.1 | 68.1 | 50.7 | 50.5 |
| **ReKnoS** | **81.1** | **71.9** | **58.5** | **52.9** |

Table 8: Performance comparison of Hits@1 and F1 scores on the WebQSP, GrailQA, and CWQ datasets.

| Model | WebQSP | | GrailQA | | CWQ | |
|---|---|---|---|---|---|---|
| | Hits@1 | F1 | Hits@1 | F1 | Hits@1 | F1 |
| ToG | 76.2 | 64.3 | 68.7 | 66.6 | 57.4 | 54.5 |
| KG-Agent | 79.2 | 77.1 | 68.9 | 65.3 | 56.1 | 52.6 |
| **ReKnoS** | **81.1** | **79.5** | **71.9** | **70.2** | **58.5** | **56.8** |

**Experimental Setup.**  While our primary focus is on reasoning methods specifically tailored for large language models (LLMs), we recognize the value of including subgraph-based reasoning baselines to provide a more comprehensive evaluation. Unlike **ReKnoS**, which leverages super-relations to enhance retrieval and reasoning efficiency for LLMs, subgraph-based approaches operate with different objectives. For instance, **UniKGQA** employs RoBERTa as its encoder, whereas LLM-based methods such as ToG utilize powerful models like GPT-3.5. Despite these fundamental differences, we report the performance of these approaches on four benchmark datasets: WebQSP, GrailQA, CWQ, and SimpleQA. For SR and UniKGQA, we follow the experimental settings in the original papers, and we use GPT-3.5 for our framework.

**Results and Discussion.**  Table 7 presents the comparison between ReKnoS and the subgraph-based reasoning methods. Our method achieves superior performance across all datasets, demonstrating the effectiveness of leveraging super-relations in KGQA. Notably, **ReKnoS** outperforms **UniKGQA** by **6.0%**, **3.8%**, **7.8%**, and **2.4%** on WebQSP, GrailQA, CWQ, and SimpleQA, respectively while outperforming **SR** by even more. These results reinforce our argument that reasoning methods designed for LLMs can effectively enhance knowledge graph-based question answering.

### D.3    ANALYSIS OF HITS@1 VS. F1 METRIC

To comprehensively evaluate the effectiveness of our proposed **ReKnoS** framework, we compare the use of the Hits@1 and F1 metrics in knowledge graph question answering (KGQA) tasks.

**Motivation.**  While Hits@1 is commonly used in KGQA benchmarks, the F1 metric may provide additional insights, especially when multiple valid answers exist for a given question. However, we follow the existing work on ToG (Sun et al., 2023) in primarily reporting Hits@1, as certain datasets in our evaluation, such as **T-REx** and **zsRE**, do not involve multiple correct answers per question, making the F1 metric less meaningful in these cases. For single-answer tasks, Hits@1 remains the most reliable evaluation metric.

**Experimental Results.**  Despite the above considerations, we acknowledge the value of the F1 metric in datasets where multiple answers exist. Thus, we report both Hits@1 and F1 scores for the state-of-the-art models across three datasets: **WebQSP**, **GrailQA**, and **CWQ**. The results are summarized in Table 8. We use GPT-3.5 for this experiment.

**Discussion.**  From the results in Table 8, we observe that a stronger baseline like KG-Agent does not always maintain better F1 scores. Interestingly, the relatively weaker baseline ToG can demonstrate a more complete answer set, which is useful in scenarios where answer coverage is more critical than strict correctness. Notably, **ReKnoS** consistently outperforms both baselines, achieving the best

Table 9: Comparison of ReKnoS with additional baselines.

| Model | WebQSP | | GrailQA | |
|---|---|---|---|---|
| | Hits@1 | F1 | Hits@1 | F1 |
| RoG | 80.0 | 70.8 | 57.8 | 56.2 |
| GNN-RAG | 82.8 | 73.5 | 62.8 | 60.4 |
| ReKnoS | 81.1 | 79.5 | 58.5 | 56.8 |

Table 10: Distribution of non-retrieval cases across datasets.

| Dataset | Path Absence | Depth Limit | Misdirection |
|---|---|---|---|
| GrailQA | 6.0% | 25.2% | 68.8% |
| CWQ | 9.2% | 43.1% | 47.7% |
| WebQSP | 11.6% | 15.9% | 72.5% |

performance across both the Hits@1 and F1 metrics, demonstrating its ability to effectively balance precision and completeness.

These findings highlight that while Hits@1 is appropriate for single-answer tasks, F1 can be an important complementary metric for tasks with multiple correct answers.

### D.4 COMPARISON WITH ADDITIONAL BASELINES

In this subsection, we consider additional baselines that are applicable to the task that we focus on. These methods provide interesting perspectives.

- RoG (Luo et al., 2024): RoG is a reasoning framework that enhances LLMs with knowledge graphs by using a planning-retrieval-reasoning approach, where relation paths generated from KGs guide the retrieval of valid reasoning paths, enabling faithful, interpretable, and state-of-the-art KG reasoning. This method uses LLaMA2-Chat-7B as the LLM backbone, fine-tuned on the training split of WebQSP, CWQ, and Freebase for 3 epochs.
- GNN-RAG (Mavromatis & Karypis, 2024): GNN-RAG is a novel RAG framework that combines the reasoning capabilities of Graph Neural Networks (GNNs) with the language understanding abilities of LLMs by extracting KG reasoning paths from a dense subgraph and verbalizing them for LLM-based KGQA. This method uses LLaMA2-Chat-7B as the LLM backbone.

We also considered EWEK-QA (Dehghan et al., 2024), which is an enhanced citation-based QA system that improves knowledge extraction by combining an adaptive Web retriever with efficient knowledge graph (KG) integration. However, this method primarily focuses on citation-based question answering, which diverges from the scope of our work. Additionally, its performance has been reported in Jiang et al. (2024) to be inferior to the baselines that we included. Therefore, we did not include this method in the experiments.

From the results presented in Table 9, we observe that GNN-RAG generally achieves better results than **ReKnoS** on both datasets. This demonstrates the importance of exploiting the structural information in knowledge graphs to extract useful knowledge. Nevertheless, despite the strong performance of RoG and GNN-RAG, they require fine-tuning on white-box LLMs, which can be infeasible in specific scenarios where computational resources are scarce. On the other hand, the advantage of **ReKnoS** is that any black-box LLM can be easily plugged in to the framework.

### D.5 BREAKDOWN OF NON-RETRIEVAL CASES ON ADDITIONAL DATASETS

In this subsection, we include the breakdown of non-retrieval cases across two additional datasets: WebQSP and CWQ. The results are shown in Table 10. From the results, we observe that the reasons for non-retrieval cases are distributed differently across datasets. However, the majority of cases still fall under *Misdirection*, which **ReKnoS** is designed to address.

Table 11: Evaluation results on datasets without using built-in super-relations.

| Method | CWQ | WebQSP | OBQA | MetaQA |
|---|---|---|---|---|
| StructGPT | 55.2 | 75.2 | 77.2 | 81.2 |
| ToG | 56.2 | 76.4 | 81.0 | 87.2 |
| ReKnoS (without given super-relations) | **57.6** | **76.9** | **85.3** | **92.5** |

Table 12: Comparison of StructGPT implementations with different numbers of selected relations per step. We include the results for ToG and ReKnoS for reference.

| Model | WebQSP | | | GrailQA | | |
|---|---|---|---|---|---|---|
| | # Calls | Time | Hits@1 | # Calls | Time | Hits@1 |
| ToG | 15.3 | 4.9 | 76.2 | 18.9 | 6.8 | 68.7 |
| StructGPT (3 Relations) | 11.0 | 2.0 | 75.2 | 13.3 | 3.5 | 66.4 |
| StructGPT (1 Relation) | 3.8 | 1.1 | 72.7 | 4.2 | 1.0 | 61.7 |
| ReKnoS | 7.2 | 3.7 | 81.1 | 10.2 | 5.1 | 71.9 |

### D.6 APPLICABILITY BEYOND THE WIKIDATA KNOWLEDGE GRAPH

To further demonstrate the robustness of our method, we conducted additional experiments on two datasets with different knowledge graphs:

**OBQA (OpenBookQA)** (Mihaylov et al., 2018) is a multiple-choice question-answering task designed to assess commonsense knowledge. The utilized KG is ConceptNet (Speer et al., 2017), a comprehensive knowledge base covering diverse aspects of everyday knowledge, such as *<"rabbit", CapableOf, "move fast">*.

**MetaQA (MoviE Text Audio QA)** (Zhang et al., 2018) is a dataset focused on the movie domain, featuring questions where the answer entities can be located up to three hops from the topic entities within a movie KG derived from OMDb.

Both OBQA and MetaQA do not have built-in super-relations. In addition, we removed predefined super-relations from CWQ and WebQSP. The results are shown in Table 11.

The results validate that our approach is effective even in scenarios where super-relations must be derived dynamically via clustering, without reliance on predefined super-relations. This further reinforces the generalizability and robustness of **ReKnoS** across diverse KG structures.

### D.7 ANALYSIS OF LLM CALLS IN STRUCTGPT

In this subsection, we modified the StructGPT implementation by modifying the number of selected relations per step. The original StructGPT implementation selects only one relation at a time, whereas our modified implementation selects three relations per step, aligning with ToG and ReKnoS. This modification allows StructGPT to explore a broader set of candidate relations, thereby improving accuracy (Hits@1). However, it also increases the number of LLM calls and query time compared to its original implementation.

To provide a clearer comparison, we present an analysis of the trade-offs between the original StructGPT implementation (selecting 1 relation) and our modified implementation (selecting 3 relations). The results are summarized in Table 12.

The results illustrate the trade-offs in StructGPT. When selecting only one relation per step, the method reduces query time and the number of LLM calls, but at the cost of slightly lower accuracy (Hits@1). Conversely, when selecting three relations, StructGPT achieves higher accuracy but requires more calls and query time.

Table 13: Performance comparison with and without explicit scoring criteria.

| Model | WebQSP | GrailQA | CWQ |
|---|---|---|---|
| ReKnoS | 81.0 | 71.2 | 57.7 |
| ReKnoS (Explicit Scoring Criteria) | **81.9** | **72.5** | **58.2** |

## D.8 BENEFITS OF EXPLICIT CRITERIA

In this subsection, we integrate explicit scoring criteria to enhance the interoperability of our framework. We replace the manual example with the following scoring criteria, which serve as a guideline for the LLMs:

- **[0.8 – 1.0]: Highly Relevant**
    - There is a **strong logical connection** between the relation and the query.
- **[0.6 – 0.8]: Strongly Related**
    - The relation aligns well with the query and provides **substantial relevant information**, but it may lack some precision.
- **[0.4 – 0.6]: Moderately Related**
    - The relation is **partially relevant** and may provide **general background information** or indirect support for the query.
- **[0.2 – 0.4]: Weakly Related**
    - The relation has a **tenuous connection** to the query, providing **limited or peripheral information**.
- **[0.0 – 0.2]: Irrelevant**
    - The relation is **unrelated** or only tangentially related to the query.

By providing this guideline as an additional input to the LLM, we conduct further experiments and obtain the results shown in Table 13. The results indicate that incorporating explicit scoring criteria improves performance across all three datasets. This suggests that a well-defined guideline can enhance the model's ability to distinguish between different levels of relevance, thereby improving both interpretability and accuracy.

## D.9 MITIGATING HALLUCINATION

In this subsection, we discuss the mitigation of the risk of hallucination in our framework. Our framework ensures that all selected super-relations exist within the knowledge graph. First, the candidate selection step explicitly filters super-relations to include only those that are directly connected to previously selected relations in the KG. This design ensures that the reasoning process remains grounded in the structure of the KG and prevents the hallucination of non-existent relations from influencing the retrieval process. Second, during the scoring step, the LLM is tasked with ranking super-relations for their relevance, but these candidates are pre-selected based on their existence and connectivity in the KG. As a result, any hallucinated suggestions by the LLM are naturally excluded from the retrieval pipeline.

