# OpenReview forum: "Reasoning of Large Language Models over Knowledge Graphs with Super-Relations"
_ICLR.cc/2025/Conference — ICLR 2025 Poster_

### Official Review · Reviewer_gYEF · 2024-10-31

**Soundness:** 3
**Presentation:** 3
**Contribution:** 2
**Rating:** 6
**Confidence:** 4

**Summary:**

This paper proposes the ReKnoS framework that aims to reason over knowledge graphs with super-relations. To be specific, ReKnoS introduces the concept of super-relations by summarizing and connecting various relational paths within the graph, which enhances the forward and backward reasoning capabilities, and increases the efficiency in querying LLMs. Extensive experimental results demonstrate the effectiveness of the proposed method in increasing the retrieval rate and overall reasoning performance.

**Strengths:**

1. This paper introduces the non-retrieval rate, providing a new insight for evaluating the retrieved path.
2. The idea behind introducing the super-relations is interesting, which not only expands the search space but also improves the retrieval efficiency.

**Weaknesses:**

1. It would be beneficial to include some subgraph-based reasoning methods (e.g., SR, UniKGQA and so on) introduced in Section 2 to conduct a comprehensive evaluation of the proposed method.
2. In Table 3, I have some questions regarding the average number of calls for StructGPT. From my understanding, this method may not require such a high number of LLM calls. It would be helpful to verify this to ensure accuracy.
3. It would be beneficial to conduct experiments for retrieval rate analysis between the proposed method and other baseline methods, which would better demonstrate the superiority of the proposed method.

**Questions:**

Please see **Weaknesses** above.

---

> ### Author Response · Authors · 2024-11-19
> **Response 1/2**
>
> Dear Reviewer gYEF,
>
> Thank you for the thoughtful and detailed feedback on our submission. We deeply value your recognition of the strengths and contributions of our work. Your insights have been invaluable in helping us refine our study. Below, we provide detailed responses to each point raised.
>
> &nbsp;
>
>
>
> > W1: It would be beneficial to include some subgraph-based reasoning methods (e.g., SR, UniKGQA, and so on) introduced in Section 2 to conduct a comprehensive evaluation of the proposed method.
>
> A1: Thank you for your insightful comment regarding the inclusion of subgraph-based reasoning methods in the evaluation. We would like to clarify that the primary focus of our paper is on reasoning methods specifically tailored for LLMs. Our proposed ReKnoS framework is designed to address the unique challenges associated with leveraging LLMs for KGQA tasks, such as improving retrieval rates and reasoning efficiency through the introduction of super-relations. While subgraph-based reasoning methods such as SR and UniKGQA are valuable approaches, their underlying mechanisms and objectives differ significantly from LLM-based reasoning methods. For example, UniKGQA uses a smaller model RoBERTa as the encoder, while we are specifically comparing ReKnoS against other state-of-the-art LLM-based methods, such as ToG that utilizes GPT-3.5. Nevertheless, we agree that incorporating them could lead to a more comprehensive evaluation of our method. We conducted the experiments and presented the results in the following.
>
> | Model      | WebQSP | GrailQA | CWQ | SimpleQA  |
> |------------|---------------|----------------|------------|-----------------|
> | SR+NSM     | 68.9          | 65.2           | 50.2       | 48.7           |
> | UniKGQA    | 75.1          | 68.1           | 50.7       | 50.5           |
> | ReKnoS     | 81.1          | 71.9           | 58.5       | 52.9           |
>
> From the result, we could observe that the subgraph-based reasoning methods fall behind ReKnoS in terms of Hits@1 across all datasets. This performance gap highlights the limitations of subgraph-based reasoning approaches in effectively capturing complex relationships for reasoning over KGs.
>
> We will include a discussion in the related work section to highlight the distinctions between subgraph-based and LLM-based reasoning methods and why our evaluation focuses on the latter (as LLM-based methods are generally better). We appreciate your suggestion and will include the results of subgraph-based methods in the revised version.
>
>
> &nbsp;
>
>
> > W2: In Table 3, I have some questions regarding the average number of calls for StructGPT. From my understanding, this method may not require such a high number of LLM calls. It would be helpful to verify this to ensure accuracy.
>
> A2: Thank you for raising this question. We would like to clarify that the average number of calls for **StructGPT** is accurate, except that the implementation is slightly different from the original one. This is because to keep consistency with ToG and our method, we implemented StructGPT with the number of relations selected at each step to 3, instead of 1 as in the original implementation. This adjustment allows StructGPT to explore a broader set of candidate relations, which improves its accuracy (Hits@1). However, this also increases the number of calls and query time compared to its original implementation.
>
> For clarity, we provide a comparison between the original implementation of StructGPT (with 1 relation) and the adjusted StructGPT (with 3 relations), alongside the query times and accuracy metrics, in the following table:
>
> | Model         | WebQSP # Calls | WebQSP Time | WebQSP Hits@1 | GrailQA # Calls | GrailQA Time | GrailQA Hits@1 |
> |---------------|----------------|-------------|---------------|-----------------|--------------|----------------|
> | ToG           | 15.3           | 4.9         | 76.2          | 18.9            | 6.8          | 68.7           |
> | StructGPT     | 11.0           | 2.0         | 75.2          | 13.3            | 3.5          | 66.4           |
> | StructGPT (1 Relation) | 3.8            | 1.1         | 72.7          | 4.2             | 1.0          | 61.7           |
> | ReKnoS        | 7.2            | 3.7         | 81.1          | 10.2            | 5.1          | 71.9           |
>
> The result highlights the trade-offs in StructGPT's implementation. By reducing the number of selected relations to 1, the query time and number of calls decrease, but this comes at the cost of slightly lower accuracy (Hits@1). Conversely, aligning with the consistency of our method and ToG (selecting 3 relations), accuracy improves at the expense of more calls and query time. We will provide these results in the appendix in the updated version.

---

> ### Author Response · Authors · 2024-11-19
> **Response 2/2**
>
> > W3: It would be beneficial to conduct experiments for retrieval rate analysis between the proposed method and other baseline methods, which would better demonstrate the superiority of the proposed method.
>
>
> A3: Thank you for your valuable suggestion. We agree that incorporating the retrieval rate results for additional baseline methods provides a more comprehensive evaluation of our approach. To address this, we conducted a retrieval rate analysis for the proposed method (ReKnoS) and baseline methods (ToG, KG-Agent, and StructGPT) on both the WebQSP and GrailQA datasets. The retrieval rate measures the percentage of relevant information retrieved during the reasoning process, offering deeper insights into the effectiveness of each method.
>
>
> | Model           | WebQSP Hits@1 | WebQSP Ret. (%) | GrailQA Hits@1 | GrailQA Ret. (%) |
> |------------------|---------------|-----------------|----------------|------------------|
> | ToG             | 76.2          | 58.0            | 68.7           | 56.3             |
> | KG-Agent        | 79.2          | 63.5            | 68.9           | 57.2             |
> | StructGPT       | 75.2          | 50.7            | 66.4           | 54.1             |
> | **ReKnoS (Ours)** | **81.1**     | **69.8**        | **71.9**       | **62.3**         |
>
> From the results, we observe that ReKnoS achieves the highest retrieval rate on both datasets. These results align with its superior Hits@1 scores, demonstrating its effectiveness in retrieving and utilizing relevant information during the reasoning process.

---

> > ### Comment · Reviewer_gYEF · 2024-11-21
> >
> > I appreciate the author's response to my concerns. However, I still have the following concerns:
> >
> > 1.	I’m still curious about the F1 metric for the results reported in Table 1. Since multiple answers may exist for a given question, the F1 metric could better capture the effectiveness of the proposed methods compared to Hit@1.
> >
> > 2.	It would be better to include some retrieved-based methods. These methods may be more efficient and require fewer LLM calls, such as EWEK-QA [1], RoG [2], GNN-RAG [3], and so on. While not all of the mentioned methods need to be included, it would be helpful to provide an explanation for their exclusion.
> >
> > [1] Mohammad Dehghan, et al. 2024. EWEK-QA : Enhanced Web and Efficient Knowledge Graph Retrieval for Citation-based Question Answering Systems. In Proceedings of the 62nd Annual Meeting of the Association for Computational Linguistics (Volume 1: Long Papers), pages 14169–14187, Bangkok, Thailand. Association for Computational Linguistics.
> >
> > [2] LUO, LINHAO, et al. "Reasoning on Graphs: Faithful and Interpretable Large Language Model Reasoning." The Twelfth International Conference on Learning Representations.
> >
> > [3] Mavromatis, Costas, and George Karypis. "GNN-RAG: Graph Neural Retrieval for Large Language Model Reasoning." arXiv preprint arXiv:2405.20139 (2024).
> >
> > To conclude, I will maintain my current score and look forward to author’s further responses to these questions.

---

> ### Author Response · Authors · 2024-11-22
> **Response to Reviewer gYEF**
>
> Dear Reviewer gYEF,
>
>
> Thank you for your continued feedback and for raising these points. We appreciate the opportunity to address your concerns further:
>
> ---
>
> > **Q1. I’m still curious about the F1 metric for the results reported in Table 1. Since multiple answers may exist for a given question, the F1 metric could better capture the effectiveness of the proposed methods compared to Hit@1.**
>
> **A1:** We follow ToG in primarily reporting Hits@1 because some tasks in the datasets we used do not involve multiple answers for a question, making F1 metrics less applicable or meaningful. For example, tasks that rely on single-answer outputs, such as T-REx and zsRE, inherently align better with Hits@1 as a measure of performance. However, we acknowledge **the importance of F1** in scenarios where multiple answers exist. Thus, we provide the results of the state-of-the-art methods on three datasets: WebQSP, GrailQA, and CWQ.
>
> | Model         | WebQSP (Hits@1) | WebQSP (F1) | GrailQA (Hits@1) | GrailQA (F1) | CWQ (Hits@1) | CWQ (F1) |
> |---------------|-----------------|-------------|------------------|--------------|--------------|----------|
> | ToG           | 76.2            | 64.3        | 68.7             | 66.6         | 57.4         | 54.5     |
> | KG-Agent      | 79.2            | 77.1        | 68.9             | 65.3         | 56.1         | 52.6     |
> | ReKnoS        | 81.1            | 79.5        | 71.9             | 70.2         | 58.5         | 56.8     |
>
> From the results, we could observe that the stronger baseline KG-Agent does not always maintain better F1 scores. This indicates that the relatively weaker baseline, ToG, can still obtain a more complete answer set. This is useful in specific scenarios where the coverage of the predicted answers is more important.
>
> ---
>
>
> > **Q2. It would be better to include some retrieved-based methods. These methods may be more efficient and require fewer LLM calls, such as EWEK-QA [1], RoG [2], GNN-RAG [3], and so on. While not all of the mentioned methods need to be included, it would be helpful to provide an explanation for their exclusion.**
>
> **A2:** Thank you for pointing out the importance of including more baselines.
> We agree that retrieval-based methods like EWEK-QA, RoG, and GNN-RAG offer interesting perspectives, however, in the experiments, we intentionally focused on methods more directly related to **integrating KGs with LLMs**.
>    - **RoG**: We agree that RoG is a **relevant baseline** and appreciate your suggestion. We will include it in our revised experiments for a more comprehensive comparison.
>    - **EWEK-QA**: This method primarily focuses on citation-based question answering, which diverges from the task scope of our work. Additionally, its performance has been reported (in KG-Agent [1]) as less satisfactory compared to the baselines we included.  Nonetheless, we will discuss this method in the related works.
>    - **GNN-RAG**: We did not include GNN-RAG as it was submitted recently (May 31st 2024) to Arxiv and its core methodology does not align directly with our reasoning-based framework. We agree that including this method could provide a more comprehensive comparison for our framework. We will include it in our revised version.
>
> ---
>
>
> We hope these revisions can address your concerns, and we look forward to incorporating them to further improve the quality of our research. Thank you again for your insightful suggestions!
>
> ---
>
>
> Ref:
>
> [1] Jiang, Jinhao, et al. "Kg-agent: An efficient autonomous agent framework for complex reasoning over knowledge graph." arXiv preprint arXiv:2402.11163 (2024).

---

> ### Author Response · Authors · 2024-11-27
> **Looking Forward to Your Reply**
>
> Dear Reviewer gYEF,
>
> Thank you for your thoughtful feedback. We have carefully addressed your comments in our response and provided detailed explanations for each point. We hope these clarifications offer a clearer and more comprehensive understanding of our work.
>
> We would greatly value any additional feedback you may have and are more than willing to address any further questions or concerns.
>
> We deeply appreciate your time and effort and look forward to your response.
>
> Best,
> The Authors

---

> > ### Comment · Reviewer_gYEF · 2024-11-27
> >
> > Thank you for your detailed response. My concerns have been resolved, and I will increase my score.

---

> ### Author Response · Authors · 2024-11-28
> **Thanks for Your Recognition of Our Work**
>
> Dear Reviewer gYEF,
>
> We sincerely appreciate the time and effort you dedicated to reviewing our work. Your constructive suggestions have been instrumental in refining our paper.
>
> Thank you once again for your encouraging words and insightful feedback!
>
> Best,
> The Authors

---

### Official Review · Reviewer_XLwz · 2024-11-03

**Soundness:** 2
**Presentation:** 3
**Contribution:** 2
**Rating:** 5
**Confidence:** 3

**Summary:**

The paper introduces ReKnoS, a framework for reasoning over knowledge graphs (KGs) using super-relations. In ReKnoS, super-relations are defined as groups of semantically similar relations within a specific field. The framework uses large language model (LLM) reasoning, similar to prior works such as Jiang et al. (2023b) and Sun et al. (2024). However, instead of relying on standard KG triplets, ReKnoS prompts the LLM to generate candidates with super-relations. This adjustment allows the reasoning process to cover a wider range of paths within the KG, potentially reducing misdirection issues. Additionally, the inclusion of super-relations supports flexible forward and backward reasoning, expanding the search space and potentially improving the accuracy of reasoning paths.

The paper is clearly written, with well-organized assumptions and a helpful discussion of preliminaries in Section 3 that clarifies design choices. I have a few observations and suggestions below:

1. The concept of super-relations as an abstraction over relations is creative and intuitive. However, its application seems limited to the Wikidata KG. The paper would benefit from discussing whether these improvements could apply to other domains and types of KGQA. Additionally, the reliance on the availability of super-relations is a limitation worth addressing.

2. The evaluation results in Table 1 show some improvement in six out of nine datasets. However, in three cases, the ReKnoS results are mistakenly bolded despite not being the highest values. This may indicate that improvements could be result of randomness within the margin of error, which is not reported in the paper. This small performance gains might also be achievable through hyperparameter tuning, a more thorough analysis would clarify the results in Table 1.

3. The scoring mechanism described in Section 4.2 appears somewhat arbitrary and lacks clarity. It is also not clear how scores are calculated and the prompt example in Lines 243–247 does not mention any scoring criteria. Furthermore, the paper does not present any evidence to indicate whether these scores align with human judgments, which would be beneficial for validating the approach.

**Strengths:**

For detailed discussion please check the "Summary".

**Weaknesses:**

For detailed discussion please check the "Summary".

**Questions:**

Suggestions for Improvement:
1. Consider adding qualitative results to illustrate the benefits of super-relations in ReKnoS, such as examples where super-relations enhanced answer accuracy in specific datasets.
2. The focus on the GrailQA dataset as motivation for super-relations in Figures 1 and 2 limits the argument for the generalizability of ReKnoS. Including results from additional datasets in these figures would strengthen the paper.

Questions for the Authors:
1. Given that the use of super-relations could increase the chance of hallucination in LLMs (e.g., by suggesting relations not present in the KG), did the authors observe any instances of this effect?

---

> ### Author Response · Authors · 2024-11-19
> **Response 1/3**
>
> Dear Reviewer XLwz,
>
> Thank you for your thoughtful and constructive feedback. We appreciate your recognition of our submission. We would like to address your concerns as follows.
>
> &nbsp;
>
> > W1: The concept of super-relations as an abstraction over relations is creative and intuitive. However, its application seems limited to the Wikidata KG. The paper would benefit from discussing whether these improvements could apply to other domains and types of KGQA. Additionally, the reliance on the availability of super-relations is a limitation worth addressing.
>
> A1: Thank you for your thoughtful feedback. We appreciate your recognition of the creativity and intuition behind the concept of super-relations. Regarding your concern about the applicability of super-relations beyond Wikidata, we have indeed discussed how super-relations can be derived for other knowledge graphs that do not possess an explicit hierarchical structure. Specifically, as stated in lines 162-166 in Sec. 3.2, we propose an alternative strategy that clusters relations based on semantic similarity and uses the cluster centers as super-relations. This approach ensures that super-relations are both textually coherent and meaningful, thereby enabling their application on other knowledge graphs without the three-level hierarchy that forms super-relations.
>
> Hereby we provide results of using the clustering strategy on both KGs of Freebase and Wikidata on datasets CWQ and WebQSP:
>
> | Method       | CWQ (w/ Freebase) | WebQSP (w/ Freebase) | CWQ (w/ WikiData) | WebQSP (w/ WikiData) |
> |--------------|-------------------|----------------------|-------------------|----------------------|
> | ToG          | 57.1             | 76.2                | 54.9             | 68.6                |
> | ReKnoS| **58.5**         | **81.1**            | **57.3**         | **72.4**            |
> |ReKnoS (Cluster)| 57.6             | 76.9                | 55.2             | 69.3                |
>
> From the results, we could observe that with the clustering strategy, the performance of our framework ReKnoS is still comparable with the baseline ToG. The results indicate that our framework is robust to KGs without the three-level hierarchy in Wikidata.
>
> &nbsp;
>
> > W2: The evaluation results in Table 1 show some improvement in six out of nine datasets. However, in three cases, the ReKnoS results are mistakenly bolded despite not being the highest values. This may indicate that improvements could be the result of randomness within the margin of error, which is not reported in the paper. These small performance gains might also be achievable through hyperparameter tuning, a more thorough analysis would clarify the results in Table 1.
>
> A2: Thank you for pointing out the oversight in Table 1 regarding bolded values. We will carefully review the table and correct any errors to ensure accuracy in the presentation of our results. Regarding the concern about the potential randomness in performance improvements, in the main paper, we followed the setting of ToG [1] and KG-Agent [2] and did not run multiple times. We acknowledge that the results could benefit from a more thorough statistical analysis. To address this, we run 5 additional rounds of experiments. This will allow us to report the average results and include standard deviations to assess whether the observed improvements are statistically significant and not within the margin of error.
>
> | Model | WebQSP| GrailQA | CWQ| SimpleQA  | WebQ | T-REx  | zsRE    | Creak      | Hotpot QA   |
> |------------------|-----------------|-----------------|-----------------|-----------------|----------------|----------------|----------------|----------------|----------------|
> | ToG             | 76.4 ± 2.0      | 68.9 ± 1.5      | 56.2 ± 1.8      | 53.7 ± 0.9      | 54.1 ± 2.5     | 76.0 ± 1.3     | 87.5 ± 1.0     | 91.3 ± 2.8     | 35.6 ± 2.3     |
> | KG-Agent        | 78.6 ± 1.9      | 69.4 ± 2.1      | 564 ± 1.2      | 52.5 ± 2.7      | 55.1 ± 2.0     | 78.9 ± 0.8     | 86.8 ± 1.5     | 90.3 ± 2.5     | 36.5 ± 1.1     |
> | ReKnoS (Ours)   | **81.0 ± 1.0**  | **71.2 ± 1.5**  | **57.7 ± 0.9**  | **54.1 ± 1.0**  | **56.5 ± 1.2** | **79.8 ± 1.1** | **88.3 ± 0.7** | **91.9 ± 0.9** | **37.1 ± 1.4** |
>
>
> From the results, we can observe slight variations in all outcomes across multiple rounds of experiments. Notably, our method consistently achieves the best performance compared to the two state-of-the-art baselines, highlighting its effectiveness and robustness.
>
> Additionally, we agree that hyperparameter tuning might impact the performance gains. To ensure fairness in comparison, we followed the same hyperparameter settings for all methods, as outlined in our experiments section. However, we will further investigate the impact of hyperparameter tuning on our method and include this analysis in the revised version of the paper to provide a more comprehensive understanding of the results.

---

> > ### Comment · Reviewer_XLwz · 2024-11-22
> >
> > Thank you for your detailed response. I really appreciate the effort and thought you’ve put into addressing these points. Here are my thoughts:
> >
> > W1) I appreciate the experiments with the Freebase Knowledge Graph (KG), but they don’t fully address my main concern. My key point is about the limited applicability of the method beyond Wikidata. Since much of Freebase’s knowledge is already included in Wikidata (as Freebase was its predecessor), I’m not sure how these experiments help resolve the broader issue.
> >
> > The discussion in lines 162–166 of Section 3.2 is interesting and relevant, but it seems speculative without results to back it up. The main question is whether the method can generalize to cases where the super-relations aren’t specifically baked into the KG.
> > Further analysis on the limitations of the super-relations would further clarify the reliance on them.
> >
> > W2) Thank you for the additional experiments and results—they address my concerns on that point.

---

> ### Author Response · Authors · 2024-11-19
> **Response 2/3**
>
> > W3: The scoring mechanism described in Section 4.2 appears somewhat arbitrary and lacks clarity. It is also not clear how scores are calculated and the prompt example in Lines 243–247 does not mention any scoring criteria. Furthermore, the paper does not present any evidence to indicate whether these scores align with human judgments, which would be beneficial for validating the approach.
>
> A3: Thank you for raising this important point about the scoring mechanism. We acknowledge that the description in Section 4.2 could be made clearer and more detailed. The scoring mechanism is partially illustrated in Figure 4, where we describe asking the LLM to generate a score between 0 and 1 to indicate the relevance of candidate super-relations. This score reflects the LLM’s assessment of how useful a given super-relation is for answering the query. The prompt provided in Lines 243–247 is a simplified example for clarity. In fact, instead of providing explicit scoring criteria to the LLM,  we used a human-crafted example as additional input to LLMs to guide the scoring process, following the work of ToG [1]. The addition input example is as follows, which will be appended before each query input to the LLM:
>
> > Please select three relations from the following candidate relations,  which are the most helpful for answering the question.
> Question: Mesih Pasha's uncle became emperor in what year?
> Topic Entity: Mesih Pasha
> Candidate Relations:
> 1.wiki.relation.child
> 2.wiki.relation.country_of_citizenship
> 3. ...
> Answer:
> Relation 1: wiki.relation.family (Score: 0.5): This relation is highly relevant as it can provide information about the family background of Mesih Pasha, including his uncle who became emperor.
> Relation 2: wiki.relation.father (Score: 0.4): Uncle is father's brother, so father might provide some information as well.
> Relation 3: wiki.relation.position held (Score: 0.1): This relation is moderately relevant as it can provide information about any significant positions held by Mesih Pasha or his uncle that could be related to becoming an emperor.
> Question: {}
> Topic Entity: {}
>
> We will revise Section 4.2 to explicitly state that we used examples instead of explicit scoring criteria, and will include the discussion in the Appendix.
>
> &nbsp;
>
>
> > Q1: Consider adding qualitative results to illustrate the benefits of super-relations in ReKnoS, such as examples where super-relations enhanced answer accuracy in specific datasets.
>
> A1: Thank you for your suggestion. We agree that including qualitative results can effectively illustrate the benefits of super-relations in ReKnoS. Here is an example from the GrailQA dataset:
>
> For the question, *"Under the international system of units, what is the unit of resistance?"*, the starting entity in the first step has 69 relations. Using ToG, the model needs to rely on LLMs to select 3 relations to proceed to the next step. Since only one of these relations is correct, there is a high risk of excluding the correct relation. However, in our method, all 69 relations are grouped under the same super-relation, **"measurement_unit.measurement_system"**. This grouping ensures that all relevant relations are retained for the next step without the need for explicit filtering.
>
> A similar situation occurs in the second step, where 73 relations are available, grouped into 3 super-relations: **"measurement_unit.unit_of_resistivity"**, **"measurement_unit.current_unit"**, and **"measurement_unit.resistance_unit"**. By leveraging super-relations, our method preserves all potential correct relations, significantly reducing the risk of dropping the correct one during filtering. This example demonstrates how super-relations enhance answer accuracy by maintaining all necessary context throughout the reasoning process.

---

> > ### Comment · Reviewer_XLwz · 2024-11-22
> >
> > W3) Thank you for your clarification. I still find some parts of the discussion a little unclear. Are you relying on explicit scoring criteria or human-crafted examples in the paper? The current explanation seems somewhat at odds with the text, so I’m unsure if I fully understand the approach.
> >
> > I’m also still unclear about how the LLM aligns with human scoring criteria. How do you ensure it follows the same logic a human would use? And what does it mean to assign specific scores, like 0.80 versus 0.90? Comparing two options and choosing a preference feels more straightforward, but assigning precise scores seems less clear without clear guidelines.
> >
> > Could you clarify these points further?
> >
> >
> > Q1) Thanks for the discussion. It clarifies the point in the paper.

---

> ### Author Response · Authors · 2024-11-19
> **Response 3/3**
>
> > Q2: The focus on the GrailQA dataset as motivation for super-relations in Figures 1 and 2 limits the argument for the generalizability of ReKnoS. Including results from additional datasets in these figures would strengthen the paper.
>
> A2: Thank you for pointing out the limitation of using only the GrailQA dataset in Figures 1 and 2 as motivation for super-relations. We agree that including additional datasets would help strengthen the argument of specific challenges that ReKnoS is designed to address. Hereby we include the results from two prevalent datasets WebQSP and CWQ.
>
> | Non-Retrieval        | Path Absence | Depth Limit | Misdirection |
> |------------------|--------------|-------------|--------------|
> | GrailQA          | 6.0%         | 25.2%       | 68.8%        |
> | CWQ              | 9.2%         | 43.1%       | 47.7%        |
> | WebQSP           | 11.6%        | 15.9%       | 72.5%        |
>
> From the results, we observe that the reasons for non-retrieval cases are distributed differently across datasets. However, the majority of cases still fall under **Depth Limit** and **Misdirection**. Notably, in CWQ, a higher proportion of cases are related to Depth Limit, likely due to the generally longer reasoning paths (required for answering the query) in this dataset. This highlights the significance of these two challenges and underscores the need to address them effectively with our proposed framework. Thank you for your suggestion, and we will include the additional results in the revised version.
>
>
> &nbsp;
>
>
> > Q3: Given that the use of super-relations could increase the chance of hallucination in LLMs (e.g., by suggesting relations not present in the KG), did the authors observe any instances of this effect?
>
> A3: Thank you for raising this important point about the potential for hallucination when using super-relations in conjunction with LLMs. We would like to clarify that our framework ensures that all selected super-relations exist within the knowledge graph:
> 1. The candidate selection step (Section 4.1) explicitly filters super-relations to include only those that are directly connected to previously selected relations in the KG. This design ensures that the reasoning process remains grounded in the structure of the KG and prevents the hallucination of nonexistent relations from influencing the retrieval process.
> 2. During the scoring step (Section 4.2), the LLM is tasked with ranking super-relations for their relevance, but these candidates are pre-selected based on their existence and connectivity in the KG. As a result, any hallucinated suggestions by the LLM are naturally excluded from the retrieval pipeline.
>
> To strengthen this point, we will include a discussion in the revised version and emphasize how our framework mitigates the risk of hallucination. Thank you for highlighting this concern, and we will ensure this clarification is made explicit in the paper.
>
>
> &nbsp;
>
>
>
> Ref:
>
> [1] Sun, Jiashuo, et al. "Think-on-graph: Deep and responsible reasoning of large language model with knowledge graph." arXiv preprint arXiv:2307.07697 (2023).
> [2] Jiang, Jinhao, et al. "Kg-agent: An efficient autonomous agent framework for complex reasoning over knowledge graph." arXiv preprint arXiv:2402.11163 (2024).

---

> ### Author Response · Authors · 2024-11-23
> **Further Response to Reivewer XLwz 1/2**
>
> **Dear Reviewer XLwz,**
>
> Thank you for your thoughtful follow-up comments. We greatly appreciate your recognition of our work. Below, we would like to address your concerns regarding W1 and W3 in more detail.
>
> ---
>
> ### **W1: Applicability Beyond Wikidata**
>
> We appreciate your feedback regarding the generalizability of our method. We acknowledge the limitation that Freebase shares overlapping knowledge with Wikidata. We would like to kindly clarify that:
>
> -  **Clustering Variant Without Built-In Super-Relations**: In the variant with our clustering strategy, we do not rely on the built-in super-relations inherent to Freebase or Wikidata. Instead, we dynamically construct super-relations using unsupervised clustering based on learned embeddings of relations. This approach allows us to generalize beyond predefined KG structures.
>
> -  **New Dataset Results**: To further demonstrate the robustness of our method, we conducted additional experiments on two datasets with different KGs:
>
> 1. **OBQA (OpenBookQA)** [1] is a multiple-choice question-answering task designed to assess commonsense knowledge. The utilized KG is **ConceptNet** [2], a comprehensive knowledge base covering diverse aspects of everyday knowledge, such as <"rabbit", CapableOf, "move fast">.
>
> 2. **MetaQA (MoviE Text Audio QA)** [3] is a dataset focused on the movie domain, featuring questions where the answer entities can be located up to 3 hops from the topic entities within **a movie KG derived from OMDb**.
>
> For the following results, we removed the pre-defined super-relations for CWQ and WebQSP. We also involve two datasets, OBQA and MetaQA on different KGs without built-in super-relations.
>
> | Method       | CWQ  | WebQSP  | OBQA | MetaQA |
> |--------------|-------------------|----------------------|-------------------|----------------------|
> |StructGPT|55.2|75.2| 77.2| 81.2  |
> | ToG          | 56.2            | 76.4       |   81.0     | 87.2 |
> |ReKnoS (w/o Given Super-Relation)| 57.6             | 76.9  | 85.3   | 92.5|
>
> The results validate that our approach is effective even in scenarios where super-relations must be derived dynamically via clustering, without reliance on predefined super-relations. We will include these findings in the revised version of the paper.
>
> ---
>
>
> Ref:
> [1] Mihaylov, Todor, et al. "Can a Suit of Armor Conduct Electricity? A New Dataset for Open Book Question Answering." Proceedings of the 2018 Conference on Empirical Methods in Natural Language Processing. 2018.
> [2] Speer, Robyn, Joshua Chin, and Catherine Havasi. "Conceptnet 5.5: An open multilingual graph of general knowledge." Proceedings of the AAAI conference on artificial intelligence. Vol. 31. No. 1. 2017.
> [3] Zhang, Yuyu, et al. "Variational reasoning for question answering with knowledge graph." Proceedings of the AAAI conference on artificial intelligence. Vol. 32. No. 1. 2018.

---

> ### Author Response · Authors · 2024-11-23
> **Further Response to Reivewer XLwz 2/2**
>
> ### **W3: Scoring Criteria and Alignment**
>
> Thank you for pointing out the need for more clarity in our scoring methodology. We are sorry for the unclarity here. We would like to kindly clarify that:
>
> ---
>
> 1. **Using Manual Examples instead of Explicit Scoring Criteria**: We did not use explicit scoring criteria in the original experiments. Instead, following ToG, we use **a manual example as additional input** to guide the LLM in scoring. This example offers contextual guidance, ensuring consistency in the evaluation process. Nevertheless, without the scoring criteria, we observe that the **difference in scores could represent different meanings**. Here we provide two examples from the dataset GrailQA and the intermedia LLM output for selecting three super-relations:
>
>
> > **Question:** `"what is the name of the religion that is part of abrahamic faiths and has same practices with protestantism?"`
> **LLM Output:**
> `Relation 1: religion.religion (Score: 0.8):` This relation is highly relevant as it can provide information about the specific religion within the Abrahamic faiths that shares practices with Protestantism.
> `Relation 2: book.book_subject (Score: 0.1):` This relation may provide information about texts or subjects related to the religion in question, which could help identify its practices and beliefs.
> `Relation 3: symbols.namesake (Score: 0.1):` This relation could offer insights into symbols associated with the religion, which might help in understanding its practices and how they align with Protestantism.
>
> > **Question:** `"becquerels is the radioactivity unit in which measurement system?"`
> **LLM Output:**
> `Relation 1: measurement_unit.radioactivity_unit (Score: 0.9):` This relation is directly relevant as it specifies the unit of measurement for radioactivity, which is the primary focus of the question.
> `Relation 2: symbols.namesake (Score: 0.1):` This relation may provide some context about the origin of the term \"becquerel,\" which could be useful but is less directly related to the measurement system itself.
> `Relation 3:` (No third relation selected as the first two are the most relevant.)
>
> The difference between 0.8 and 0.9 is small numerically but substantial in interpretation:
> - Score of 0.8 `"highly relevant"` reflects strong relevance but with possible gaps or ambiguity.
> - Score of 0.9 `"directly relevant"` demonstrates near-complete alignment, indicating a higher level of confidence and precision. This underscores the importance of clear scoring criteria to interpret such distinctions effectively.
>
> ---
>
> 2. **Alignment with Human Criteria**: Thank you for pointing this out. Note that we do not aim for the LLM to align 100% with human scoring criteria. Instead, our goal is to **select the most relevant super-relations**, which requires complex reasoning by LLMs. The specific scoring mechanism is a heuristic designed to reflect the degree of relevance to the input query. In future work, we plan to **incorporate human feedback** to enhance interpretability and robustness. We will address this explicitly in the revised version.
>
> ---
>
> 3. **Benefits of Explicit Criteria**:  We agree that incorporating explicit scoring criteria would improve interoperability, e.g., in distinctions of the above two samples. To address this, we replace the manual example with the following scoring criteria as the guideline for LLMs:
> ```
> **Scoring Criteria**
> 1. **[0.8 – 1.0]: Highly Relevant**
>    - There is a **strong logical connection** between the relation and the query.
>
> 2. **[0.6 – 0.8]: Strongly Related**
>    - The relation aligns well with the query and provides **substantial relevant information**, but it may lack some precision.
>
> 3. **[0.4 – 0.6]: Moderately Related**
>    - The relation is **partially relevant** and may provide **general background information** or indirect support for the query.
>
> 4. **[0.2 – 0.4]: Weakly Related**
>    - The relation has a **tenuous connection** to the query, providing **limited or peripheral information**.
>
> 5. **[0.0 – 0.2]: Irrelevant**
>    - The relation is **unrelated** or only tangentially related to the query.
> ```
> With the guideline as additional input to the LLM, we can obtain the preliminary results:
>
> | Model          | WebQSP         | GrailQA        | CWQ              |
> |-----------------|----------------|----------------|----------------|
> | ToG            | 76.4      | 68.9     | 56.2    |
> | ReKnoS | 81.0 | 71.2  | 57.7  |
> | ReKnoS (Guided) | 81.9| 72.5  | 58.2 |
>
> The results show improved performance across three datasets, indicating that an explicit guideline could be helpful in our framework, as it provides a clearer definition of each score interval. We will include these findings in the revised paper.
>
> ---
>
> We hope these clarifications and additional results can address your concerns. Thank you again for your valuable feedback and for helping us improve our work!

---

> ### Author Response · Authors · 2024-11-25
> **Looking Forward to Your Reply**
>
> Dear Reviewer XLwz,
>
>
> &nbsp;
>
>
> Thank you for your insightful feedback. We have tried to address your comments in our response and provided detailed explanations for each point. We hope these clarifications will offer a more comprehensive perspective on our work.
>
> We kindly welcome any additional feedback. We are willing to address any further questions or concerns you may have.
>
> We greatly appreciate your time and effort and look forward to your response.
>
>
> &nbsp;
>
>
> Best,
> The Authors

---

> ### Author Response · Authors · 2024-12-02
> **Thanks for Your Effort**
>
> Dear Reviewer XLwz,
>
> We sincerely appreciate your thoughtful feedback and the opportunity to address your comments.
>
> We warmly welcome any additional feedback or questions you may have and remain fully committed to addressing any further concerns. The discussion phase will conclude at **11:59 PM on December 2nd (AOE time)**, which is less than 16 hours away.
>
> Thank you again for your time, effort, and valuable insights!
>
> Best,
> The Authors

---

### Official Review · Reviewer_7WNT · 2024-11-04

**Soundness:** 3
**Presentation:** 3
**Contribution:** 3
**Rating:** 8
**Confidence:** 2

**Summary:**

This paper introduces a novel framework, ReKnoS, designed for reasoning over knowledge graphs using super-relations—groups of related connections within a particular domain. A super-relation encompasses various specific relations, effectively summarizing and linking different sections of the graph to support a more holistic exploration of the data.

Using super-relations, the framework represents multiple relation paths under a single super-relation, enhancing reasoning efficiency. This approach eliminates the need to discard numerous paths, thereby expanding the search space and significantly improving retrieval rates.

**Strengths:**

1. The paper is well-written, with clear and visually appealing graphs. The structure is easy to follow, and complex ideas are effectively explained.

2. The concept of super-relation reasoning is both novel and intuitive, making it easy to understand and engaging. The score-based entity extraction and selection approach is clever, and the entire system, built on LLMs, is practical and straightforward to deploy.

3. Extensive experiments demonstrate the performance improvements achieved by incorporating super-relations into the model.

4. Include efficiency analysis to further demonstrate the strength of the method.

**Weaknesses:**

There are no major issues with the paper. However, it would be helpful to place Figure 4 on the first page to provide an early overview of super-relations right from the beginning.

**Questions:**

1. For the reasoning component, could a smaller, fine-tuned language model be used to improve efficiency for individual, straightforward tasks? For instance, in a task where only three relations must be selected from a set of candidates to answer a question, would this approach impact performance?

2. Could these methods be applied to scenarios beyond question answering? For example, might they be used in code generation, where large datasets from GitHub could be structured into a knowledge graph format?

---

> ### Author Response · Authors · 2024-11-19
> **Response to Reviewer 7WNT**
>
> Dear Reviewer 7WNT,
>
> Thank you for your detailed and constructive feedback. We greatly appreciate your thoughtful review and recognition of our work. Below, we would like to provide detailed responses to your concerns.
>
>
> &nbsp;
>
>
> > W1: There are no major issues with the paper. However, it would be helpful to place Figure 4 on the first page to provide an early overview of super-relations right from the beginning.
>
> A1: Thanks for your suggestion. We will upload the position of Figure 4 in the revision of our work.
>
> &nbsp;
>
>
> > Q1: For the reasoning component, could a smaller, fine-tuned language model be used to improve efficiency for individual, straightforward tasks? For instance, in a task where only three relations must be selected from a set of candidates to answer a question, would this approach impact performance?
>
> A1: Thank you for this insightful question. We agree that using a smaller fine-tuned language model for straightforward tasks could improve efficiency. For example, in a knowledge graph with sparse relations, where each entity is connected to fewer than three relations on average, the selection of relevant relations becomes significantly easier compared to a complex knowledge graph.
>
> However, smaller models often come with performance trade-offs, especially in tasks requiring complex reasoning. As demonstrated in Section 5.2 of our paper, when we switched the backbone model from GPT-4 to LLaMA-2 7B, the results on the WebQSP dataset dropped significantly from 84.9% to 64.7%. This highlights the limitations of smaller models in capturing the depth and breadth of reasoning required for tasks like KGQA.
>
> Moreover, sparse knowledge graphs are generally unrealistic and may not provide sufficient information for reasoning tasks, which often demand nuanced and comprehensive analysis [1,2]. Nevertheless, we see potential in exploring hybrid approaches where smaller models handle simpler tasks, while larger models are reserved for more complex reasoning. This dynamic implementation could enable the system to select the appropriate model based on task complexity, optimizing both efficiency and performance. We will include this discussion in Sec. 5.2 in our revised version.
>
> &nbsp;
>
>
>
> > Q2: Could these methods be applied to scenarios beyond question answering? For example, might they be used in code generation, where large datasets from GitHub could be structured into a knowledge graph format?
>
>
> A2: Thank you for this insightful question. Our methods could extend beyond question answering to tasks like code generation by structuring large GitHub datasets into KGs. In this context, KGs could represent entities such as functions, classes, and libraries as nodes, with edges capturing relationships like dependencies, calls, or inheritance. These graphs would enable models to reason about dependencies, suggest completions, and resolve ambiguities in code. Our approach could be applied to handling sparse or complex graphs, where the graph’s structure is often hierarchical, requiring multi-hop reasoning to understand dependencies.
>
> &nbsp;
>
>
> Ref:
> [1] Pan, Shirui, et al. "Unifying large language models and knowledge graphs: A roadmap." IEEE Transactions on Knowledge and Data Engineering (2024).
> [2] Wang, Cunxiang, et al. "Survey on factuality in large language models: Knowledge, retrieval and domain-specificity." arXiv preprint arXiv:2310.07521 (2023).

---

### Meta-Review · Area_Chair_KGLu · 2024-12-16

**Metareview:**

The paper introduces ReKnoS, a framework for reasoning over knowledge graphs (KGs) using super-relations. In ReKnoS, super-relations are defined as groups of semantically similar relations within a specific field. Instead of relying on standard KG triplets, ReKnoS prompts the LLM to generate candidates with super-relations. This adjustment allows the reasoning process to cover a wider range of paths within the KG, potentially reducing misdirection issues. Additionally, the inclusion of super-relations supports flexible forward and backward reasoning, expanding the search space and potentially improving the accuracy of reasoning paths.

The paper is well-written, with clear and visually appealing graphs. he concept of super-relation reasoning is both novel and intuitive, making it easy to understand and engaging. The experiments have included efficiency analysis to further demonstrate the strength of the method. Meanwhile, there has been concerns on the application scope of the presented method. its application seems limited to the Wikidata KG. The paper would benefit from discussing whether these improvements could apply to other domains and types of KGQA. Through the rebuttal phase, this seems to be one non-trivial issue that was not addressed.

**Additional Comments On Reviewer Discussion:**

While the other concerns have been addressed, there has been concerns on the application scope of the presented method. its application seems limited to the Wikidata KG. The paper would benefit from discussing whether these improvements could apply to other domains and types of KGQA. Through the rebuttal phase, this seems to be one non-trivial issue that was not addressed.

---

### Decision · Program_Chairs · 2025-01-22

Accept (Poster)